# MAKING SLOW THINKING FASTER: COMPRESSING LLM CHAIN-OF-THOUGHT VIA STEP ENTROPY

**Zeju Li**[1], **Jianyuan Zhong**[1], **Ziyang Zheng**[1], **Xiangyu Wen**[1], **Zhijian Xu**[1], **Yingying Cheng**[2],
**Fan Zhang**[2], **Qiang Xu**[1,3*]

[1]The Chinese University of Hong Kong
[2]Huawei Technologies Co., Ltd
[3]Shenzhen Loop Area Institute

```
{zjli24, jyzhong, zyzheng23, xywen22, zjxu21, qxu}@cse.cuhk.edu.hk
{cheng.yingying1, zhang.fan2}@huawei.com
```

## ABSTRACT

Large Language Models (LLMs) using Chain-of-Thought (CoT) prompting excel at complex reasoning but generate verbose thought processes with considerable redundancy, leading to increased inference costs and reduced efficiency. We introduce a novel CoT compression framework based on **step entropy**, a metric that quantifies *the informational contribution of individual reasoning steps* to identify redundancy. Through theoretical analysis and extensive empirical validation on mathematical reasoning benchmarks, we demonstrate that steps with low entropy are indeed highly redundant. Our experiments reveal that an astonishing 80% of low-entropy intermediate steps can be pruned with minor degradation in the final answer accuracy across DeepSeek-R1-7B, 14B and Qwen3-8B. This finding sharply contrasts with random or high-entropy pruning, which severely impairs reasoning performance. Building on this, we propose a novel two-stage training strategy combining Supervised Fine-Tuning (SFT) and Group Relative Policy Optimization (GRPO) reinforcement learning. This approach enables LLMs to autonomously learn to generate compressed COTs during inference by *strategically incorporating [SKIP] tokens*. Our method significantly improves LLM inference efficiency while preserving accuracy, paving the way for more scalable LLM deployments and a better understanding of their internal reasoning. The code and data are released in `https://github.com/staymylove/COT_Compresstion_via_Step_entropy`.

## 1 INTRODUCTION

Large Language Models (LLMs) have demonstrated remarkable capabilities in complex reasoning tasks, particularly when employing techniques like Chain-of-Thought (COT) (Wei et al., 2022). By generating explicit intermediate reasoning steps, often referred to as "slow thinking", Large Reasoning Model (LRM) such as the DeepSeek-R1 (Guo et al., 2025) Series and Qwen3 (Yang et al., 2025) significantly enhance performance on multi-step problems in domains like mathematics, coding and symbolic logic. This process allows the model to break down complex problems into more manageable components, leading to more reliable and accurate outcomes.

However, a notable drawback of current slow thinking COT implementations is the considerable redundancy often present within the generated thought processes (Deng et al., 2023; Zhong et al., 2025a). These verbose reasoning paths, while thorough, lead to increased inference latency, higher computational costs, and diminished overall efficiency. As models become larger and are deployed at scale, these inefficiencies present a significant bottleneck for practical applications.

To mitigate this, prior research has explored several compression strategies. One prominent direction focuses on making the CoT process implicit or latent, finetuning the model to internalize reasoning

---

*Corresponding Author

steps without verbalizing them (Deng et al., 2024; Hao et al., 2024) or dynamically compressing them in latent space (Tan et al., 2025). Other work has focused on compressing the reasoning chain at different granularities, from pruning tokens in the input context (Li et al., 2023), enabling controllable token-level skipping during generation (Xia et al., 2025), to chunk-based compression (Wang et al., 2025a). While these methods improve efficiency, they do not offer a principled way to identify and remove entire reasoning steps that are semantically redundant.

Intuitively, when humans tackle complex problems, they record only key milestones, omitting obvious thoughts. Recent work has sought to teach LLMs a similar ability to skip steps (Liu et al., 2024; Jiang et al., 2025) or tune for length-compressible CoTs (Ma et al., 2025b). However, a fundamental question persists: how can we systematically identify which steps in a reasoning chain are crucial versus superfluous?

In this paper, we propose *a novel, entropy-based* method to identify and quantify the significance of each step within an LLM's Chain-of-Thought. We introduce the concept of **step entropy**, a metric that measures the informational contribution of individual reasoning steps by aggregating token-level entropy during generation. Our core hypothesis is that steps with lower entropy represent more predictable, and therefore less informative, parts of the reasoning chain that can be safely pruned without compromising accuracy.

To validate this approach, we conduct systematic empirical analysis by calculating step entropy for reasoning trajectories and investigating the impact of pruning varying proportions of steps (10% to 100%) using three strategies: low-entropy pruning, high-entropy pruning, and random pruning. Our findings, as shown in Figure 1, reveal that pruning up to 80% of low-entropy steps maintains accuracy while achieving substantial token reductions (16-45% across multiple benchmarks), whereas high-entropy step removal causes immediate performance degradation. Cross-model validation on DeepSeek-R1 (7B, 14B) and Qwen3-8B demonstrates the universality of our entropy-based approach across different architectures. To demonstrate the superiority of step-level pruning, we compare our method with direct token-level pruning in the experimental section.

Building on this validation, we introduce a two-stage training strategy combining Supervised Fine-Tuning (SFT) with Group Relative Policy Optimization (GRPO) (Shao et al., 2024) that enables models to autonomously generate compressed reasoning trajectories. The SFT stage teaches models to predict when to use [SKIP] tokens based on entropy-compressed training data, while GRPO optimizes a composite reward function balancing accuracy, compression ratio, and response length. Our trained models achieve 35-57% token reductions while maintaining or improving accuracy, demonstrating that LLMs can learn to perform efficient reasoning without sacrificing quality.

The main contributions of our work are summarized as follows:

- We introduce step entropy as a principled metric for quantifying the contribution of each step in the Chain-of-Thought thinking trajectory.

- We provide strong empirical evidence that low-entropy steps are largely redundant and can be pruned up to 80% without significant loss of accuracy.

- We propose a two-stage training strategy that enables LLM to learn the efficient compressed reasoning policy, significantly improving inference efficiency while maintaining performance.

## 2 RELATED WORK

### 2.1 LLM REASONING WITH REINFORCEMENT LEARNING

Reinforcement Learning (RL) has emerged as a powerful paradigm for enhancing the reasoning capabilities of Large Language Models (LLMs). Recent advancements, such as those demonstrated in Shao et al. (2024) and Xie et al. (2025), showcase RL's efficacy in refining LLMs' ability to tackle complex reasoning tasks. Furthermore, strategies involving "long COT" Yeo et al. (2025) and "slow thinking" Zhong et al. (2025b) (which involves extending inference time) Comanici et al. (2025); Guo et al. (2025); OpenAI (2025) have been shown to significantly improve LLM reasoning performance by allowing for more elaborate and deliberate thought processes.

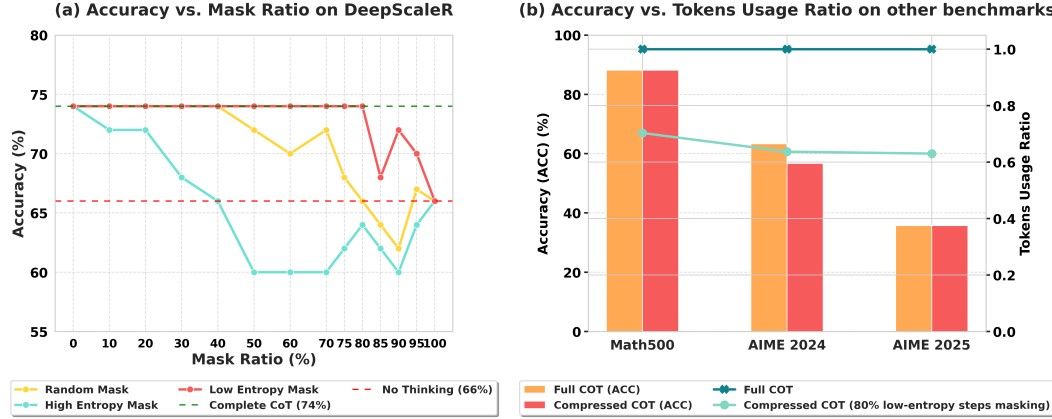

Figure 1: Comprehensive Performance of COT Compression via Step Entropy. (a) Accuracy vs. Mask Ratio on 50 samples from DeepScaleR. This plot illustrates the impact of different pruning strategies (Random, High-Entropy Steps, Low-Entropy Steps) on final answer accuracy as the mask ratio of intermediate COT steps increases. Note that pruning up to **80% low-entropy steps** maintains Complete COT accuracy (b) Accuracy vs. Tokens Usage Ratio on other benchmarks. This plot compares the accuracy and token usage ratio of the Full COT against our Compressed COT (80% low-entropy steps pruning) across Math500, AIME 2024, and AIME 2025 on DeepSeek-R1-7B.

However, the increased length and computational overhead associated with these verbose COTs have led to concerns regarding efficiency. Wang et al. (2025b); Cuadron et al. (2025); Sui et al. (2025) highlight the phenomenon of "overthinking," where excessively long COTs can paradoxically lead to diminished efficiency without proportional gains in accuracy. This emphasizes the need for methods that can optimize the length and content of reasoning trajectories.

## 2.2 COT COMPRESSION AND LATENT REASONING

To address the inefficiency of verbose reasoning, researchers have pursued two main avenues: compressing the explicit CoT and making the reasoning process entirely implicit.

Explicit compression methods aim to shorten the generated text at various granularities. At the finest level, some works enable controllable token-level skipping (Xia et al., 2025) or explore the information-theoretic minimum number of tokens required for a solution (Lee et al., 2025). At a coarser grain, R1-Compress introduces chunk-based compression and search (Wang et al., 2025a). Other strategies use length-constrained tuning, integrating penalties into RL reward functions (Shen et al., 2025; Hou et al., 2025) or developing specific architectures for length-compressible CoTs like CoT-Valve (Ma et al., 2025b). Our work advances this line by proposing a more semantically grounded approach: we operate at the step level, arguing it better mimics human cognition (skipping entire thoughts, not words). Furthermore, our method is guided by an explicit information-theoretic signal—step entropy—teaching the model not just to be shorter, but to selectively discard what is verifiably uninformative.

An alternative, more radical approach is to make reasoning implicit or latent. Methods like iCOT (Deng et al., 2024) and COCONUT (Hao et al., 2024) fine-tune models to internalize reasoning steps, while others use knowledge distillation to embed the process in the model's hidden states (Deng et al., 2023). More recently, dynamic latent compression performs reasoning entirely within these hidden states, avoiding explicit generation altogether (Tan et al., 2025). While these latent strategies offer maximum efficiency, they sacrifice the critical interpretability and verifiability of an explicit CoT. Our work carves a distinct path by focusing on optimizing the explicit reasoning chain, preserving its benefits while drastically improving its efficiency.

## 3 CoT Compression via Step Entropy

This section details our framework for CoT compression, which is built on a novel, entropy-based metric. We begin by formally defining step entropy and providing its theoretical justification as a measure of a reasoning step's importance. We then describe our primary contribution: the low-entropy steps pruning strategy and the process for performing LLM inference with the compressed CoT.

### 3.1 Step Entropy

The foundational premise of our work is that not all steps in a CoT contribute equally to the final answer. To formalize this, we introduce *step entropy* as a measure of the informational contribution of each reasoning step. We hypothesize that steps generated with high confidence (low uncertainty) by the model are more likely to be redundant. Information entropy provides a natural way to quantify this uncertainty.

Given a CoT sequence generated by a LRM, we first segment it into a series of distinct steps, $C = (S_1, S_2, \ldots, S_N)$, where each step $S_i$ is a sequence of tokens, $S_i = (t_{i,1}, t_{i,2}, \ldots, t_{i,M_i})$. During autoregressive generation, for each token $t_{i,j}$, the model produces a probability distribution $p(\cdot|c_{i,j})$ over its entire vocabulary $V$, where $c_{i,j}$ is the context consisting of the input prompt and all previously generated tokens.

The entropy of this distribution, which represents the model's uncertainty at that generation step, is calculated using the standard Shannon entropy formula:

$$H(t_{i,j}|c_{i,j}) = - \sum_{w \in V} p(w|c_{i,j}) \log_2 p(w|c_{i,j}) \tag{1}$$

We define the **step entropy** $H(S_i|S_{<i})$ as the sum of token-level entropy across all tokens within that step:

$$H(S_i|S_{<i}) = \sum_{j=1}^{M_i} H(t_{i,j}|c_{i,j}) = H(t_{i,1}, ..., t_{i,M_i}|S_{<i}) \tag{2}$$

A high step entropy $H(S_i)$ indicates that the model was, on average, highly uncertain when generating step $S_i$, while a low step entropy indicates the deterministic generation.

**Lemma 1** (Entropy-Bounded Information). *Given a reasoning process where a sequence of steps $C = (S_1, S_2, \ldots, S_N)$ leads to a final answer $A$, the conditional mutual information $I(S_j; A|\bar{S}_j)$ between the step and the answer, conditioned on all other steps $\bar{S}_j = C \setminus \{S_j\}$, is bounded by the step entropy $H(S_j|S_{<j})$ of the step itself. Specifically:*

$$I(S_j; A|\bar{S}_j) \leq H(S_j|S_{<j})$$

*The formal proof is provided in Appendix B.1.* This result demonstrate that when the entropy of step $S_j$ is low, the conditional mutual information of $S_j$ and the final answer $A$, i.e. the relation of $S_j$ and $A$ is minor.

**Theorem 1** (Entropy-Bounded Information on Subset). *For any subset of $K + 1$ steps $\tilde{S} = \{S_{k_0}, S_{k_1}, \ldots, S_{k_K}\} \subseteq C$, the conditional information contribution of this subset to the final answer is bounded by the sum of the step entropies of the steps within it:*

$$I(\tilde{S}; A|C \setminus \tilde{S}) \leq \sum_{i=0}^{K} H(S_{k_i}|S_{<k_i})$$

*The formal proof is provided in Appendix B.2* The result denotes that steps $S_{k_0}, S_{k_1}, ..., S_{k_K}$ could have minor relation to the final solution $A$. This observation implies that, a step with low entropy suggesting the deterministic thinking, which has minor relation to the finally solution, thus denotes such step could be less informative, and potentially redundant.

While step entropy theoretically serves as an upper bound on the mutual information between the CoT steps and the response, in practice, it suffers from bias introduced by step length. To mitigate this issue, we propose to use the *length-normalized step entropy*, as defined in Definition 1.

**Definition 1** (Length-normalized Step Entropy). *Given a reasoning step $S_i$ consisting of $M_i$ tokens and the preceding context $S_{<i}$, we define the length-normalized step entropy as:*

$$H(S_i \mid S_{<i}) = \frac{1}{M_i} \sum_{j=1}^{M_i} H(t_{i,j} \mid c_{i,j}) \tag{3}$$

*where $t_{i,j}$ represents the $j$-th token of step $i$ and $c_{i,j}$ represents the context up to that token.*

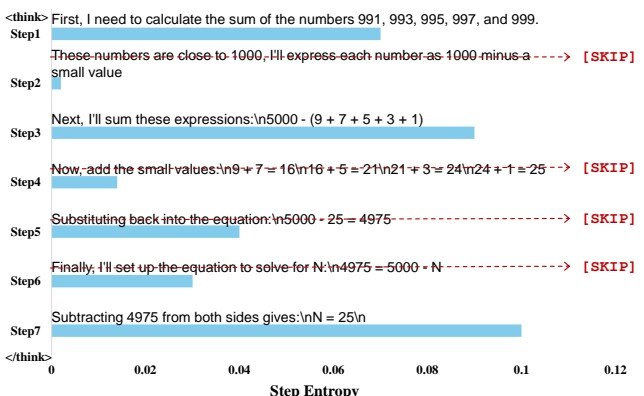

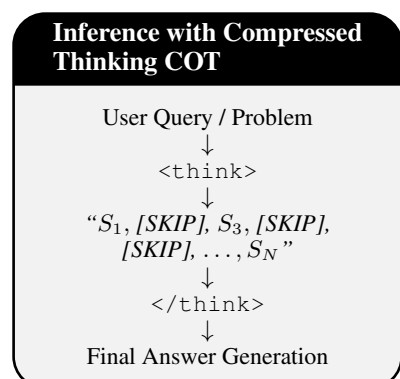

(a) The case of Low-Entropy steps pruning strategy for COT compression, and replacing each selected low-entropy step with a special '[SKIP]' token.

(b) The compressed CoT sequence (`<think>`...`</think>`) is concatenated with the user query in the prompt context.

## 3.2 LOW-ENTROPY STEPS PRUNING STRATEGY FOR COT COMPRESSION

Based on the observation established above, we propose a practical CoT compression approach that selectively removes low-entropy steps while preserving the essential reasoning structure. Our method operates on the principle that steps with low entropy are more likely to be redundant and can be safely pruned without significantly impacting the final answer quality.

1. **Generate Full CoT:** For each problem instance $x$, we use DeepSeek-R1-Distill-Qwen-7B to generate a complete CoT trajectory, the response format is `<think>` $C$ `</think>` `final answer`. The reasoning steps $S_1, S_2, \ldots, S_N$, delimited by `\n\n`, are extracted from thinking content $C$ between the `<think>` and `</think>` tags, $C = (S_1, S_2, \ldots, S_N)$.

2. **Calculate Step Entropy:** For each step $S_i \in C$, we compute its step entropy $H(S_i)$ using Equation 2.

3. **Entropy-Based Pruning:** We rank all steps in ascending order of their entropy scores and identify the $\kappa \times N$ lowest-entropy steps for pruning, where $\kappa$ is the pruning ratio hyperparameter. The compressed CoT $C'$ is constructed by replacing each selected low-entropy step with a special '[SKIP]' token, while preserving high-entropy steps in their original form. The example of this process is shown in Figure 2a.

4. **Inference with Compressed CoT:** The compressed sequence $C'$ is concatenated with the user query and the `</think>` delimiter to prompt the model to generate only the final answer, as illustrated in Figure 2b.

This approach allows us to systematically compress CoT sequences while maintaining reasoning coherence. The pruning ratio $\kappa$ provides a flexible control mechanism for balancing compression efficiency and answer quality, with optimal values determined empirically across different datasets and problem types. The upper bound of steps pruning ratio $\kappa$ will be discussed in Experiment 4.1.

## 3.3 AUTONOMOUS COMPRESSION VIA TWO-STAGE TRAINING

While our entropy-based pruning strategy effectively compresses existing CoT sequences, enabling models to autonomously generate compressed reasoning trajectories during inference represents a

more practical advancement. Our two-stage training methodology successfully achieves this goal by teaching models to balance accuracy with efficiency through learning when to skip redundant reasoning steps.

**Stage 1: Supervised Fine-Tuning (SFT)** We first train the model on (problem, compressed CoT, answer) pairs of the dataset in the Experiment 4.2 using our low entropy-based pruning strategy. The model learns to predict compressed reasoning paths and generate [SKIP] tokens by minimizing cross-entropy loss, providing robust initialization for reinforcement learning.

**Stage 2: Group Relative Policy Optimization (GRPO)** While SFT teaches static imitation of compressed traces, it does not explicitly optimize the accuracy-efficiency trade-off. We employ Group Relative Policy Optimization (GRPO)(Shao et al., 2024) to further optimize this behavior through reward-driven learning.

For each input prompt, we sample a group of $K$ completions. The model's goal is to learn a policy $\pi_\theta$ that maximizes a composite reward function $R(C)$ for each generated completion $C$. The total reward is the sum of four components designed to balance correctness with efficiency:

$$R(C) = [R_{\text{correctness}}, R_{\text{skip\_ratio}}, R_{\text{skip\_num}}, R_{\text{response\_length}}] \tag{4}$$

Let $T_{\text{think}}(C)$ be the thinking content within the completion $C$. The reward components are defined as follows:

1. **Correctness Reward ($R_{\textbf{correctness}}$):** A large positive reward for generating the correct final answer. Let $A_{\text{extracted}}(C)$ be the answer extracted from completion $C$ and $A^*$ be the ground truth.
$$R_{\text{correctness}}(C, A^*) = \begin{cases} 2.0 & \text{if } A^* == A_{\text{extracted}}(C) \\ 0.0 & \text{otherwise} \end{cases} \tag{5}$$

2. **Skip Ratio Reward ($R_{\textbf{skip\_ratio}}$):** A tiered reward for achieving a high ratio of skipped steps, encouraging compression. Let $N_{\text{skip}}$ be the count of '[SKIP]' tokens and $N_{\text{steps}}$ be the total number of steps in $T_{\text{think}}(C)$. The skip ratio is $\text{Ratio}_{\text{skip}} = N_{\text{skip}} / \max(1, N_{\text{steps}})$.
$$R_{\text{skip\_ratio}}(C) = \begin{cases} 1.0 & \text{if } \text{Ratio}_{\text{skip}} \geq \kappa_{high} \\ 0.5 & \text{if } \kappa_{low} \leq \text{Ratio}_{\text{skip}} < \kappa_{high} \\ 0.0 & \text{otherwise} \end{cases} \tag{6}$$

3. **Skip Number Penalty ($R_{sn}$):** Penalty -1.0 when [SKIP] tokens exceed $\tau_{skip\_num}$ to prevent degenerate behavior.

4. **Response Length Penalty ($R_{rl}$):** Penalty -1.0 for responses exceeding $\tau_{length}$ tokens to encourage conciseness.

This two-stage process trains the model to strategically decide when to perform detailed reasoning versus when to skip steps, achieving efficient reasoning while preserving accuracy. The experiments of training and evaluation results are presented in Section 4.3, with an ablation study of the reward components available in Appendix F.

## 4 EXPERIMENTS

This section presents experiments validating our entropy-based CoT compression method. We first conduct controlled studies to establish an optimal pruning ratio. We then demonstrate the effectiveness and generalizability of this strategy across multiple benchmarks and model sizes, justifying our step-level approach over token-level alternatives. Finally, we evaluate a two-stage training method that enables a model to autonomously generate compressed reasoning trajectories.

### 4.1 DETERMINING THE OPTIMAL PRUNING RATIO

To identify the safe threshold for step pruning, we conduct a controlled experiment using 50 samples from DeepScaleR (Luo et al., 2025) dataset on DeepSeek-R1-7B. And we investigate the impact on

Table 1: Comparing the full COT baseline with our proposed step-entropy based pruning (Our) method, which prunes 80% of the lowest-entropy steps for DeepSeek-R1-7B, 14B and Qwen-8B. We conduct experiments to get the Pass@1 Accuracy(ACC)(%) and the number of **Average Thinking Tokens per answer** (contains the Unicode characters) during the inference on GSM8k, Math500, AIME2024 and AIME2025.

| Method | GSM8k | | Math500 | | AIME 2024 | | AIME 2025 | |
|---|---|---|---|---|---|---|---|---|
| | ACC (%) | Avg Tokens | ACC (%) | Avg Tokens | ACC (%) | Avg Tokens | ACC (%) | Avg Tokens |
| DeepSeek-R1-7B | 78.54 | 298.33 | 88.17 | 3703.83 | 63.33 | 15843.43 | 35.71 | 18203.23 |
| DeepSeek-R1-7B (Our) | 80.82 | 294.29 (↓1.3%) | 88.17 | 2604.23 (↓29.7%) | 56.67 | 10092.80 (↓36.3%) | 35.71 | 11471.17 (↓37.0%) |
| DeepSeek-R1-14B | 82.64 | 283.63 | 84.37 | 2853.73 | 65.52 | 15414.83 | 58.62 | 18000.10 |
| DeepSeek-R1-14B (Our) | 84.00 | 278.29 (↓1.9%) | 82.16 | 1980.97 (↓30.6%) | 58.62 | 8705.57 (↓43.5%) | 51.72 | 10842.07 (↓39.8%) |
| Qwen3-8B | 94.46 | 3053.67 | 91.37 | 7138.49 | 79.31 | 20936.57 | 76.92 | 19902.43 |
| Qwen3-8B (Our) | 94.39 | 2557.47 (↓16.2%) | 91.13 | 5209.24 (↓27.0%) | 81.48 | 11533.57 (↓44.9%) | 76.00 | 11716.63 (↓41.1%) |

final answer accuracy by pruning steps using three distinct strategies with a mask ratio varying from 10% to 100% ("no-thinking" mode (Ma et al., 2025a)), shown in Figure 1 (left). In our methodology, pruned steps are replaced with a special "[SKIP]" token. This choice is informed by an ablation study detailed in Appendix E, which confirms that an explicit placeholder is more robust at high compression ratios than alternatives like direct removal.

- **Low-Entropy Steps Pruning:** Steps with the lowest entropy are replaced by "[SKIP]".

- **High-Entropy Steps Pruning:** Steps with the highest entropy are replaced by "[SKIP]".

- **Random Steps Pruning:** Steps are replaced by "[SKIP]". at random, serving as a control.

The results, illustrated in Figure 1, strongly validate our hypothesis. With **Low-Entropy Steps Pruning**, we observe that final answer accuracy remains stable and unaffected even when up to 80% of the lowest-entropy steps are masked. Beyond this 80% threshold, accuracy begins to decline, eventually converging to the accuracy of the "no-thinking" mode. This provides powerful evidence that a vast majority of low-entropy steps are indeed redundant. We found that with the best pruning strategy, we can prune up to 80% lowest-entropy steps (40% tokens redundancy) when not affecting the accuracy, shown in Figure 1 (right).

Conversely, with High-Entropy Steps Pruning, accuracy degrades immediately upon masking even a small fraction of steps. When the mask ratio exceeds 40%, performance drops below that of the "no-thinking" mode, indicating that removing these critical, high-information steps is more detrimental than providing no reasoning at all. The **Random Steps Pruning** strategy's performance falls between the two, beginning to decline at a 40% ratio.

Based on these findings, we establish our core strategy: pruning 80% of steps with the lowest entropy ($\kappa = 0.8$), replacing them with [SKIP] tokens while preserving the remaining high-entropy steps. Moreover, we validate this strategy also work on Deepseek-R1-14B and Qwen-8B.

## 4.2 VALIDATING LOW-ENTROPY STEPS PRUNING STRATEGY

With the 80% threshold established, we conduct extensive experiments to validate our strategy's effectiveness, generalizability, and scalability across different models and datasets.

**Models and Datasets.** We use models from the DeepSeek-R1 series (7B and 14B) and Qwen3-8B (Yang et al., 2025), which are open-source Large Reasoning Models with strong mathematical reasoning capabilities. To generate the initial CoT trajectories for our training data, we merged 40k samples from the DeepScaleR dataset (Luo et al., 2025) with 90k samples from the OpenR1-Math dataset (OpenR1-Math, 2025), creating a final composite dataset of 130k instances. To test the effectiveness and generalizability of our compression method, we evaluate performance on several standard mathematical benchmarks: GSM8k (Cobbe et al., 2021), Math500 (Lightman et al., 2023), AIME2024 (dataset card AIME, 2024) and AIME2025 (dataset card AIME, 2025).

**Performance and Generalizability on Benchmarks.** To test the broader effectiveness of our strategy, we applied the 80% low-entropy steps pruning strategy to multiple benchmarks across both Deepseek-R1-7B, 14B and Qwen3-8B models.

Table 1 presents comprehensive results comparing our compressed CoT method against full CoT baselines. Our approach consistently achieves substantial efficiency gains while maintaining or improving accuracy across all models. The DeepSeek-R1 series shows remarkable token reductions: 29.7-37.0% for the 7B model and 30.6-43.5% for the 14B model across mathematical benchmarks, with GSM8k showing slight accuracy improvements for both sizes. Notably, Qwen3-8B demonstrates the strongest performance with impressive token reductions of 16.2-44.9% while maintaining competitive accuracy and even achieving slight improvements on AIME 2024 (79.31%→81.48%). The cross-architecture consistency—spanning both DeepSeek-R1 and Qwen3 model families—demonstrates that step entropy is a robust and generalizable principle for identifying redundancy, independent of model architecture, size. For a detailed analysis of our method's performance across different domains, we conducted experiments on the MMLU benchmark (Hendrycks et al.). The results, presented in Appendix G, demonstrate that step entropy is a generalizable metric for identifying redundancy beyond mathematical reasoning.

**Scalability and Dataset Creation.** To ensure our method scales beyond controlled experiments, we further validated that the 80% low-entropy pruning strategy holds at a much larger scale. To validate this strategy at scale, we applied this compression pipeline to the entire DeepScaler (40k) and OpenR1-Math datasets (90k). The results in Table 2 confirm that even across these tens of thousands of examples, the accuracy of the statically compressed CoTs remains almost identical to that of the full CoTs. This large-scale validation not only proves the robustness of our method but also serves as the direct procedure for Dataset Creation. The final dataset (130K), consisting of (problem, compressed CoT, answer) pairs our training strategy.

Table 2: Comparing the Accuracy (%) of Full CoT (Full Chain of Thought) against Our Compressed CoT, based DeepSeek-R1-7B on two large datasets: DeepScaleR (40K) and OpenR1-Math (90K).

| Model | DeepScaleR | OpenR1-Math |
|---|---|---|
| Full CoT | 63.74 | 48.39 |
| Compressed COT | 62.17 | 47.31 |

**Discussion of Our Method v.s. Directly Masking Tokens** A crucial aspect of our methodology is the decision to prune entire reasoning steps rather than individual tokens. To justify this, we compared our step-based pruning approach against a token-based pruning baseline, where we remove the lowest-entropy tokens from the thinking trace irrespective of the steps they belong to. The results, shown in Figure 3, are unequivocal. While our step-pruning method maintains baseline accuracy even after removing up to 40% of the total thinking tokens, the token-pruning approach leads to a sharp and immediate decline in performance. Accuracy drops significantly after just a 20% token mask ratio. This demonstrates that a reasoning step is the correct semantic unit for compression. Removing individual low-entropy tokens (e.g., common words or operators) can break the syntactic and semantic integrity of a critical reasoning step, rendering it incomprehensible to the model. In contrast, removing an entire low-entropy step preserves the structure of the remaining, more important steps, leading to a much more robust compression strategy.

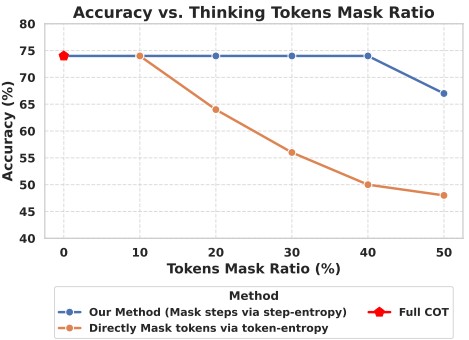

Figure 3: Comparing the accuracy of Our Method (via step-entropy) and Directly Masking Tokens (via token-entropy) across various thinking token mask ratios, with Full COT serving as the baseline of Deepseek-R1-14B on DeepScaleR dataset.

### 4.3 TRAINING AND EVALUATION FOR AUTONOMOUS COMPRESSION

We evaluate our two-stage training framework (detailed in Section 3.3) for autonomous CoT compression. We analyze the trained model's balance of accuracy and efficiency, and benchmark its performance against advanced compression baselines to demonstrate its state-of-the-art capabilities.

**Experimental Setup**   We use DeepSeek-R1-Distill-Qwen-7B with 130k mathematical problems (DeepScaleR, OpenR1-Math) preprocessed using 80% entropy-based pruning, yielding 70k training samples after filtering sequences exceeding 4096 tokens. **Stage 1 (SFT):** 3 epochs on 70k samples using DeepSpeed Stage 2 and LoRA (r=16, $\alpha$=16). **Stage 2 (GRPO):** 10k samples with reward parameters $\tau_{high} = 0.8$, $\tau_{low} = 0.5$, $\tau_{skip} = 100$, $\tau_{length} = 3500$. Training uses DeepSpeed Stage 3, LoRA (r=16, $\alpha$=32), AdamW, G=14, KL=0.04 on 8×80GB GPUs. More details and the descriptions of baselines can be found in Appendix C.

**Training Experimental Analysis**   The results of our two-stage training process, presented in Table 3, demonstrate the effectiveness of our approach in creating an efficient yet powerful LRM.

The initial **SFT stage** provides a strong foundation by teaching the model to imitate compressed reasoning paths, achieving significant token reductions (e.g., 43% on GSM8k). On AIME 2024, it achieves a 42% token reduction with a moderate drop in accuracy. The subsequent **GRPO stage** optimizes this behavior through reward-driven learning, teaching the model to intelligently balance efficiency and correctness. The final "SFT + GRPO" model achieves impressive efficiency gains across all benchmarks—a 44% token reduction on GSM8k, 35% on Math500, 57% on AIME 2024, and 41% on AIME 2025—while maintaining or even improving accuracy (e.g., GSM8k: 78.54%→79.15%). This result demonstrates that the model learns a nuanced, adaptive policy that goes beyond static imitation (see Appendix F for reward components ablation).

Finally, comparing our trained model (SFT + GRPO) to the results of static pruning inference strategy from Table 1 reveals the key advantage of our training methodology. While static pruning is effective, our trained model learns a more dynamic and ultimately superior compression policy. On the most complex benchmark, AIME 2024, our trained model not only achieves a far greater token reduction (a 57.0% drop vs. 36.3% for static pruning) but also slightly improves accuracy (57.14% vs. 56.67%). This indicates that by training the model, it learns a more sophisticated, context-aware policy that can more effectively identify and prune redundancy in complex reasoning chains than a fixed, static rule. While static pruning preserves slightly higher accuracy on simpler benchmarks like Math500, the trained model's ability to achieve state-of-the-art compression on challenging problems makes it a more robust and powerful solution for practical deployment.

Table 3: Comparison of Pass@1 Accuracy (ACC %) and Average Thinking Tokens per answer across baseline (DeepSeek-R1-7B), SFT, and SFT+GRPO training results on GSM8k, Math500, AIME2024, and AIME2025 benchmarks.

| Method | GSM8k | | Math500 | | AIME 2024 | | AIME 2025 | |
|---|---|---|---|---|---|---|---|---|
| | ACC (%) | Avg Tokens | ACC (%) | Avg Tokens | ACC (%) | Avg Tokens | ACC (%) | Avg Tokens |
| DeepSeek-R1-7B | 78.54 | 298.33 | 88.17 | 3703.83 | 63.33 | 15843.43 | 35.71 | 18203.23 |
| SFT | 78.47 | 169.65 (↓43%) | 85.92 | 2776.48 (↓25%) | 56.67 | 9231.40 (↓42%) | 30.00 | 11771.80 (↓35%) |
| SFT + GRPO | 79.15 | 168.05 (↓44%) | 85.00 | 2439.42 (↓35%) | 57.14 | 6839.37 (↓57%) | 33.33 | 10750.03 (↓41%) |

Table 4: Comparative analysis of various Chain-of-Thought (CoT) compression methods on Math500 and AIME2024 benchmarks. All values represent the percentage change in accuracy (ACC) and generated token count relative to an uncompressed Full-CoT baseline.

| Method | Math500 | | AIME 2024 | |
|---|---|---|---|---|
| | ACC | Tokens | ACC | Tokens |
| Full-CoT (Baseline) | - | - | - | - |
| Token-efficient Prompts (Lee et al., 2025) | ↓6.4% | ↓11.1% | ↓0.0 | ↑7.2% |
| LC-Prompt (Xia et al., 2025) | ↓1.6% | ↓9.8% | ↓2.6% | ↑16.5% |
| CoT-Valve (Ma et al., 2025b) | ↓10.6% | ↓48.4% | ↓15.0% | ↓34.6% |
| TokenSkip (Xia et al., 2025) | ↓5.2% | ↓11.1% | ↓12.3% | ↓27.5% |
| R1-Compress (Wang et al., 2025a) | ↓3.2% | ↓20.3% | ↓6.2% | ↓12.9% |
| Our Method (SFT + RL) | **↓3.2%** | **↓35.0%** | **↓6.2%** | **↓57.0%** |

**Comparison with Baselines**   To contextualize our performance, we compare our final trained model against several state-of-the-art CoT compression methods in Table 4. While most existing approaches offer some token savings, they often come at the cost of a significant drop in accuracy. For

instance, CoT-Valve achieves a 48.4% token reduction on Math500 but with a steep 10.6% accuracy penalty. Our method, however, establishes a new state-of-the-art in the accuracy-efficiency trade-off. On Math500, our model matches the accuracy drop of the strongest baseline, R1-Compress ($\downarrow$3.2%), while delivering nearly double the token savings ($\downarrow$35.0% vs. $\downarrow$20.3%). The advantage is even more pronounced on AIME 2024. Our method again matches the accuracy performance of R1-Compress ($\downarrow$6.2%) but achieves a massive 57.0% reduction in tokens—more than four times the savings offered by R1-Compress ($\downarrow$12.9%). This robust performance across benchmarks validates that our entropy-guided, two-stage training strategy is highly effective at producing models that reason both accurately and efficiently, outperforming existing methods.

## 5 CONCLUSION AND FUTURE WORK

**Conclusion** We introduced a novel framework using step entropy to identify and compress redundant steps in LLM Chain-of-Thought reasoning. Our empirical validation, confirmed across multiple model architectures, demonstrates that pruning up to 80% of low-entropy steps maintains accuracy while achieving substantial token reductions (16-57%). Furthermore, our two-stage training strategy successfully enables models to autonomously generate these compressed reasoning trajectories, offering significant implications for efficient LLM deployment.

**Limitations and Future Work** Our work has several limitations that suggest avenues for future research. First, our method's reliance on newline characters (\n\n) generated by LLM to segment reasoning steps and its validation primarily on mathematical and select MMLU domains may limit its generalizability. Future work should explore more robust semantic segmentation and test the step entropy framework on a broader range of open-ended and multimodal reasoning tasks. Second, the fixed 80% pruning threshold, while empirically effective, may not be universally optimal across all model architectures and problem types. Developing adaptive compression strategies that dynamically adjust the pruning ratio based on task complexity or model characteristics presents a promising direction. A more detailed discussion of these points is available in Appendix H.

## 6 ACKNOWLEDGMENTS

This work was supported in part by the CUHK Research Matching Scheme under Grant No. 7106937 and 8601130.

## ETHICS STATEMENT

We have adhered to the ICLR Code of Ethics throughout the development of this research. Our work focuses on improving the inference efficiency of large language models on publicly available mathematical and multitask benchmarks (GSM8k, Math500, AIME, MMLU, etc.). We recognize the dual-use potential of highly efficient reasoning models; while they can serve as valuable tools for education and research, they could also be misused for tasks like automated academic dishonesty. The primary goal of our research is to enhance the practical deployment of LLMs by reducing their computational cost, making powerful reasoning tools more accessible and sustainable, and to contribute to the scientific understanding of redundancy in machine-generated thought processes. Our research utilizes publicly available pre-trained models. While our two-stage training process fine-tunes these models, we have made no effort to alter or remove their inherent safety mechanisms. We believe our work on creating more efficient and transparent reasoning systems aligns with the principles of responsible AI development.

## REPRODUCIBILITY STATEMENT

We are committed to ensuring the reproducibility of our research. All experiments were conducted using publicly available large language models (DeepSeek-R1, Qwen3) and standard academic benchmarks, the specifics of which are detailed in Section 4 and the Appendix G. To facilitate the full reproduction of our findings, we will make our source code and training configurations publicly available upon publication. This release will include: (1) the implementation for calculating step entropy and performing static CoT pruning, (2) the scripts used to generate the compressed dataset for training, and (3) the complete code for both the Supervised Fine-Tuning (SFT) and Group Relative Policy Optimization (GRPO) training stages. Key hyperparameters and experimental settings, such as the pruning ratio ($\kappa$=0.8), LoRA configurations, and the specific reward components for GRPO, are described in detail in our Experimental Setup of Section 4.3 and the Training Details in the Appendix C.

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

## A  THE USE OF LARGE LANGUAGE MODELS

We use Large Language Models (LLMs), including ChatGPT and Gemini, solely for the purpose of editing and polishing the writing in this paper.

## B  PROOFS OF THEORETICAL RESULTS

### B.1  PROOF OF LEMMA 1

*Proof.* Now assume that the entropy of the step $S_j$ is low, i.e., $H(S_j|S_{<j})$ is low, we want to explore the relation of $S_j$ and final solution $A$. We consider the conditional mutual information

$$I(S_j; A|\bar{S}_j) = I(S_j; A|S_1, ..., S_{j-1}, S_{j+1}, ..., S_N) \tag{7}$$
$$= H(S_j|\bar{S}_j) - H(S_j|\bar{S}_j, A) \tag{8}$$
$$\leq H(S_j|\bar{S}_j) = H(S_j|S_{<j}, S_{>j}) \tag{9}$$
$$= H(S_j|S_{<j}) - I(S_j; S_{>j}|S_{<j}) \tag{10}$$

Since $I(S_j; S_{>j}|S_{<j}) \geq 0$, we have $I(S_j; A|\bar{S}_j) \leq H(S_j|S_{<j})$.  □

### B.2  PROOF OF THEOREM 1

*Proof.* Assume that there are K+1 steps, $\tilde{S} = S_{k_0}, S_{k_1}, ..., S_{k_K}$, and we have

$$I(\tilde{S}; A \mid (C/\tilde{S})) = \sum_{i=0}^{K} I(S_{k_i}; A|(C/[S_{k_0}, ..., S_{k_i}])) \tag{11}$$

Without loss of generality, we assume the indices $k_i$ are arranged in descending order, i.e., $k_i < k_{i-1}$. Therefore, we could split the sequence $C/[S_{k_0}, ..., S_{k_i}]$ into $S_{<k_i}$ and $S_{>k_i}/[S_{k_0}, ..., S_{k_{i-1}}]$. Now consider the item with $k_i$, according to Lemma 1, we have:

$$I(S_{k_i}; A|(C/[S_{k_0}, ..., S_{k_i}])) \leq H(S_{k_i}|C/[S_{k_0}, ..., S_{k_i}]) \tag{12}$$
$$= H(S_{k_i}|S_{<k_i}, (S_{>k_i}/[S_{k_0}, ..., S_{k_{i-1}}])) \tag{13}$$
$$\leq H(S_{k_i}|S_{<k_i}) \tag{14}$$

Therefore, we conclude that

$$I(\tilde{S}; A \mid (C/\tilde{S})) = \sum_{i=0}^{K} I(S_{k_i}; A|(C/[S_{k_0}, ..., S_{k_i}])) \tag{15}$$

$$\leq \sum_{i=0}^{K} H(S_{k_i}|S_{<k_i}) \tag{16}$$

□

## C  TRAINING DETAILS

Our experiments utilize DeepSeek-R1-7B as the base Large Reasoning Model (LRM). For data preparation, we began with an initial dataset of 130k (DeepScaleR and OpenR1-Math) mathematical problems, which were pre-processed by masking 80% of their low-entropy steps. After filtering out sequences exceeding 4096 tokens, we obtained a refined dataset of 70k samples for the initial training stage.

Stage 1: Supervised Fine-Tuning (SFT). The 70k pre-processed samples were used for SFT. This stage was conducted for 3 epochs using DeepSpeed Stage 2 for distributed training and LoRA PEFT Hu et al. (2022) with parameters r=16 and $\alpha$=16.

Stage 2: Reinforcement Learning (RL). For the RL phase, a subset of 10k data samples was randomly selected from the 70k SFT-trained samples. We employed Group Relative Policy Optimization (GRPO) to further train the SFT-initialized model. This stage also utilized DeepSpeed Stage 3 for distributed training. The optimization objective involved a composite reward function designed to balance accuracy, [SKIP] token ratio, [SKIP] token number, and overall response length. LoRA PEFT was applied with parameters r=16 and $\alpha$=32, and the AdamW optimizer Loshchilov & Hutter (2017) was used. Key GRPO parameters included G=14 samples per input and a KL coefficient of 0.04. All experiments were performed on a cluster of 8 GPUs, each has 80GB RAM and over 300 TFLOPS of BF16 compute performance.

## D  BASELINES

To rigorously evaluate our proposed method, we compare it against several representative baselines spanning different Chain-of-Thought (CoT) compression strategies. These baselines include zero-shot prompting techniques that require no additional training, as well as more advanced, state-of-the-art methods that involve fine-tuning or specialized architectures.

- Prompt-BeConcise: A zero-shot prompting technique, as explored in Lee et al. (2025), we select the prompt of `Be Concise` to encourage the model to generate a more succinct reasoning chain naturally.

- LC-Prompt (Length-Control Prompt): Another zero-shot approach, adapted from Xia et al. (2025), that directly instructs the model to reduce the length of its reasoning by a fixed proportion, using the prompt of `Please reduce about 50% of the words in the Chain-of-Thought process.`

- Advanced Compression Methods: We also compare against recent, more sophisticated techniques:

  CoT-Valve (Ma et al., 2025b): A method that fine-tunes a model to generate CoTs of a controllable length for adjusting reasoning verbosity.

  TokenSkip (Xia et al., 2025): An approach that enables controllable, token-level skipping during the generation process, allowing for fine-grained compression of the reasoning path.

  R1-Compress (Wang et al., 2025a): This method performs compression at a coarser granularity by operating on "chunks" of the reasoning chain, combining compression with a search algorithm to maintain logical coherence.

## E  ABLATION STUDY ON THE STATIC PRUNING STRATEGY

We conduct an ablation study to identify the optimal replacement strategy for pruned reasoning steps, as simply deleting them may disrupt the model's logical flow. We compare four strategies at aggressive pruning ratios (80%, 85%, and 90%) on DeepSeek-R1-7B:

- **Replace with "[SKIP]"**: Our proposed method, using a distinct special token as a placeholder.

- **Replace with "[MASK]"**: A generic mask token.

- **Replace with "and then"**: A natural language filler phrase.

- **Directly Remove Steps**: Deleting the pruned step entirely from the context.

As shown in Figure 4, all methods maintain full CoT accuracy (74%) at an 80% pruning ratio. However, as compression increases, directly removing steps or using "[MASK]" leads to faster accuracy degradation (68% at 90% pruning) compared to using explicit placeholders like "[SKIP]" or "and then" (70% at 90% pruning).

This result indicates that explicitly signaling an omitted step is critical for reasoning coherence under high compression. We selected "[SKIP]" as our primary strategy due to its superior token efficiency (a single token) and its unambiguous semantic role as a placeholder, which avoids introducing potentially confounding natural language.

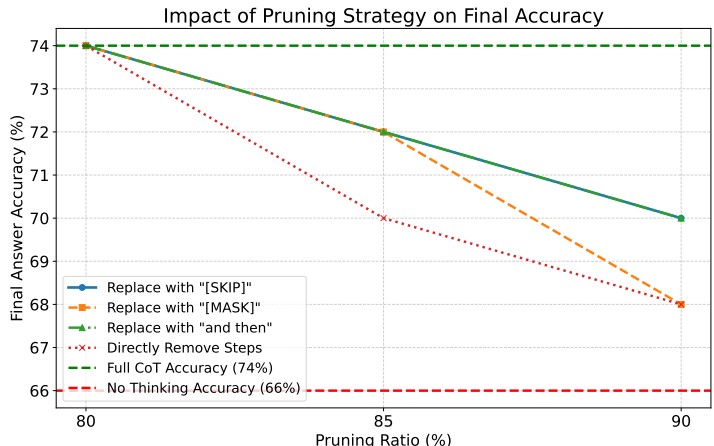

Figure 4: Ablation study on different replacement strategies for pruned low-entropy steps. The experiment is conducted on DeepSeek-R1-7B with the same sampled data of DeepScaleR in Fig 1 (left), comparing four methods for handling pruned steps at high pruning ratios (80%, 85%, 90%).

## F  REWARD COMPONENTS ABLATION STUDY

Table 5: Ablation experiment for different rewards in GRPO on GSM8k and Math500 benchmarks.

| Method | GSM8k | | Math500 | |
|---|---|---|---|---|
| | ACC (%) | Avg Tokens | ACC (%) | Avg Tokens |
| DeepSeek-R1-7B | 78.54 | 296.09 | 88.17 | 3703.83 |
| $R_c + R_{sr}$ | 75.93 | 557.17 | 51.00 | 4082.70 |
| $R_c + R_{sr} + R_{sn}$ | 78.70 | 458.27 | 85.00 | 2693.91 |
| $R_c + R_{sr} + R_{sn} + R_{rl}$ | 79.15 | 168.05 | 85.00 | 2439.42 |

Table 5 demonstrates the necessity of our multi-component reward design through systematic ablation. Using only correctness and skip ratio rewards ($R_c + R_{sr}$) yields catastrophic results, severely degrading performance (GSM8k: 78.54→75.93%, Math500: 88.17→51.00%) while paradoxically increasing token usage by 88.2% and 10.2% respectively. This indicates that naive skip optimization without constraints leads to degenerate policies generating excessive low-quality [SKIP] tokens. Adding the skip number penalty ($R_{sn}$) restores competitive accuracy but token usage remains suboptimal. The complete reward function ($R_c + R_{sr} + R_{sn} + R_{rl}$) achieves optimal balance, maintaining near-baseline accuracy while delivering substantial efficiency gains: 43.3% token reduction on GSM8k and 34.1% on Math500. These results underscore that effective CoT compression requires carefully balanced multi-objective optimization, where each component addresses specific failure modes to enable robust compression without sacrificing reasoning quality.

## G  EXTENDED EXPERIMENTS

**Model-aware Experiments**   To validate the generalizability and robustness of our step entropy-based compression method across different model architectures and scales, we conducted comprehensive model-aware experiments on four distinct Large Reasoning Models: DeepSeek-R1-7B, DeepSeek-R1-14B, Qwen3-8B, and QwQ-32B, across four mathematical reasoning benchmarks. The results in Table 6 demonstrate remarkable consistency across diverse model architectures, with our method showing consistent performance across three different model families—DeepSeek-R1, Qwen3, and QwQ. This cross-architecture consistency indicates that step entropy captures fundamental properties of reasoning redundancy that transcend specific architectural choices or training methodologies, making it a generalizable metric for identifying redundant reasoning steps. The

compression benefits scale effectively across model sizes ranging from 7B to 32B parameters, with token reduction percentages remaining relatively consistent within model families while absolute token savings increase with model scale. DeepSeek-R1 models achieve token reductions from 0.6% to 43.5% while maintaining or improving accuracy, with the 14B variant showing particularly impressive performance gains on GSM8k (82.64%→84.00%). Qwen3-8B exhibits the most aggressive compression capabilities with reductions ranging from 16.2% to 44.9%, even achieving accuracy improvements on AIME 2024 (79.31%→81.48%). QwQ-32B, as the largest model, demonstrates the highest compression potential with up to 55.1% token reduction on AIME 2024, indicating that larger models generate proportionally more redundant reasoning steps. Benchmark-specific analysis reveals distinct patterns: GSM8k shows the smallest token reductions but maintains accuracy, suggesting elementary problems require fewer redundant steps; AIME benchmarks consistently show the highest compression ratios (36.3% to 55.1%) across all models, indicating complex competition-level problems generate the most redundancy; and Math500 demonstrates balanced performance with 27.0-33.2% token reductions while maintaining high accuracy. The consistency of compression patterns across fundamentally different training methodologies provides strong evidence that step entropy captures universal properties of reasoning redundancy rather than model-specific artifacts, making our compression framework broadly applicable to current Large Reasoning Models while delivering substantial computational efficiency gains for practical deployment scenarios.

Table 6: Comparing the full COT baseline with our proposed step-entropy based pruning (Our) method, which prunes 80% of the lowest-entropy steps for DeepSeek-R1-7B, 14B, Qwen-8B and QwQ-32B. We conduct experiments to get the Pass@1 Accuracy(ACC)(%) and the number of **Average Thinking Tokens Per Answer** (contains the Unicode characters) during the inference on GSM8k, Math500, AIME2024 and AIME2025.

| Method | GSM8k | | Math500 | | AIME 2024 | | AIME 2025 | |
|---|---|---|---|---|---|---|---|---|
| | ACC (%) | Avg Tokens | ACC (%) | Avg Tokens | ACC (%) | Avg Tokens | ACC (%) | Avg Tokens |
| DeepSeek-R1-7B | 78.54 | 298.33 | 88.17 | 3703.83 | 63.33 | 15843.43 | 35.71 | 18203.23 |
| DeepSeek-R1-7B (Our) | 80.82 | 294.29 (↓1.3%) | 88.17 | 2604.23 (↓29.7%) | 56.67 | 10092.80 (↓36.3%) | 35.71 | 11471.17 (↓37.0%) |
| DeepSeek-R1-14B | 82.64 | 283.63 | 84.37 | 2853.73 | 65.52 | 15414.83 | 58.62 | 18000.10 |
| DeepSeek-R1-14B (Our) | 84.00 | 278.29 (↓1.9%) | 82.16 | 1980.97 (↓30.6%) | 58.62 | 8705.57 (↓43.5%) | 51.72 | 10842.07 (↓39.8%) |
| Qwen3-8B | 94.46 | 3053.67 | 91.37 | 7138.49 | 79.31 | 20936.57 | 76.92 | 19902.43 |
| Qwen3-8B (Our) | 94.39 | 2557.47 (↓16.2%) | 91.13 | 5209.24 (↓27.0%) | 81.48 | 11533.57 (↓44.9%) | 76.00 | 11716.63 (↓41.1%) |
| QwQ-32B | 94.09 | 1978.91 | 92.35 | 5955.39 | 79.13 | 21243.93 | 66.67 | 23711.60 |
| QwQ-32B (Our) | 93.56 | 1629.03 (↓17.7%) | 91.75 | 3978.06 (↓33.2%) | 74.07 | 9544.13 (↓55.1%) | 65.52 | 13434.17 (↓43.4%) |

Table 7: Comparing the full COT and No Thinking baseline with our proposed step-entropy based pruning method, which prunes 80% and 90% of the lowest-entropy steps for QwQ-32B. We conduct experiments to get the Pass@1 Accuracy(ACC)(%) and the number of **Average Thinking Tokens Per Answer** (contains the Unicode characters) during the inference on GSM8k, Math500, AIME2024 and AIME2025.

| Method | GSM8k | | Math500 | | AIME 2024 | | AIME 2025 | |
|---|---|---|---|---|---|---|---|---|
| | ACC (%) | Avg Tokens | ACC (%) | Avg Tokens | ACC (%) | Avg Tokens | ACC (%) | Avg Tokens |
| QwQ-32B (Full COT) | 94.09 | 1978.91 | 92.35 | 5955.39 | 79.13 | 21243.93 | 66.67 | 23711.60 |
| QwQ-32B (80%) | 93.56 | 1629.03 (↓17.7%) | 91.75 | 3978.06 (↓33.2%) | 74.07 | 9544.13 (↓55.1%) | 65.52 | 13434.17 (↓43.4%) |
| QwQ-32B (90%) | 93.56 | 1524.03 (↓23.0%) | 90.95 | 3011.36 (↓49.4%) | 70.37 | 7523.70 (↓64.6%) | 66.67 | 6459.73 (↓72.8%) |
| QwQ-32B (No thinking) | 93.86 | - | 88.53 | - | 62.07 | - | 56.67 | - |

**Domain-aware Experiments**   To evaluate the domain generalizability of our step entropy-based compression method beyond mathematical reasoning, we conducted experiments on MMLU (Massive Multitask Language Understanding) benchmarks, specifically focusing on College Medicine and High School History tasks. Tables 9 and 8 present results for DeepSeek-R1-7B and QwQ-32B models respectively, comparing different compression levels (80%, 90% low-entropy step pruning) against full CoT and no-thinking baselines. The results reveal domain-specific compression characteristics that differ significantly from mathematical reasoning tasks, with both models showing varying degrees of compression tolerance across the two MMLU domains.

DeepSeek-R1-7B demonstrates robust performance on MMLU tasks with our compression method, achieving accuracy improvements on College Medicine (61.73%→62.34%) and High School History (61.74%→64.32%) while reducing token usage by 18.6% and 7.1% respectively at 80% compression. Notably, the model maintains or improves accuracy even at 90% compression levels,

suggesting that knowledge-based reasoning tasks contain substantial redundancy that can be effectively identified through step entropy. The dramatic performance drop in the no-thinking baseline (52.46% and 47.82%) emphasizes the critical importance of maintaining some reasoning structure, validating our selective pruning approach over complete reasoning elimination.

QwQ-32B exhibits even stronger compression capabilities on MMLU benchmarks, maintaining perfect accuracy preservation on High School History (92.83%) across all compression levels while achieving substantial token reductions of up to 20.1% at 90% compression. On College Medicine, the model shows minimal accuracy degradation (86.13%→84.97%) with significant efficiency gains (15.0-20.1% token reduction). The domain-specific patterns—where History tasks show higher compression tolerance than Medicine tasks—suggest that factual recall and historical reasoning contain more redundant steps than medical reasoning, which may require more careful step-by-step analysis. These results demonstrate that our step entropy method successfully generalizes beyond mathematical domains while revealing important domain-specific characteristics that could inform adaptive compression strategies.

Table 8: QwQ-32B performance on MMLU-College Medicine and MMLU-High School History datasets showing accuracy and average tokens per answer across different pruning levels. Comparing the full COT and No Thinking baseline with our proposed step-entropy based pruning method, which prunes 80% and 90% of the lowest-entropy steps of per answer thinking tokens reduction percentages.

| Method | MMLU-College Medicine | | MMLU-High School History | |
|---|---|---|---|---|
| | ACC (%) | Avg Tokens Per Answer | ACC (%) | Avg Tokens Per Answer |
| QwQ-32B (Full COT) | 86.13 | 2912.3 | 92.83 | 1703.9 |
| QwQ-32B (80%) | 84.97 | 2475.9 (↓15.0%) | 92.83 | 1683.9 (↓1.2%) |
| QwQ-32B (90%) | 84.97 | 2326.4 (↓20.1%) | 92.83 | 1683.9 (↓1.2%) |
| QwQ-32B (No Thinking) | 84.30 | - | 91.14 | - |

Table 9: DeepSeek-R1-7B performance on MMLU-College Medicine and MMLU-High School History datasets showing accuracy and average tokens per answer across different pruning levels. Comparing the full COT and No Thinking baseline with our proposed step-entropy based pruning method, which prunes 80% and 90% of the lowest-entropy steps of per answer thinking tokens reduction percentages.

| Method | MMLU-College Medicine | | MMLU-High School History | |
|---|---|---|---|---|
| | ACC (%) | Avg Tokens Per Answer | ACC (%) | Avg Tokens Per Answer |
| DeepSeek-R1-7B (Full COT) | 61.73 | 2612.7 | 61.74 | 2054.3 |
| DeepSeek-R1-7B (80%) | 62.34 | 2127.9 (↓18.6%) | 64.32 | 1907.5 (↓7.1%) |
| DeepSeek-R1-7B (90%) | 62.34 | 2069.4 (↓20.8%) | 61.74 | 1936.2 (↓5.7%) |
| DeepSeek-R1-7B (No Thinking) | 52.46 | - | 47.82 | - |

**Key Findings and Analysis**   Our comprehensive experimental evaluation reveals several critical insights about Chain-of-Thought compression via step entropy. The most significant finding is that 80% of low-entropy reasoning steps can be safely pruned without accuracy degradation across multiple model architectures and reasoning domains. This substantial redundancy indicates that current Large Reasoning Models generate highly verbose thought processes, with the majority of steps contributing minimal informational value to final answer quality. The cross-architectural consistency of our results—spanning DeepSeek-R1 (7B, 14B), Qwen3-8B, and QwQ-32B—demonstrates that step entropy captures fundamental properties of reasoning redundancy that transcend specific model designs. Token reductions ranging from 16.2% to 55.1% across mathematical benchmarks, combined with maintained or improved accuracy, provide strong evidence that our entropy-based metric successfully identifies genuinely redundant reasoning components rather than model-specific artifacts. Domain-specific compression patterns emerge from our MMLU experiments, revealing that factual reasoning tasks (High School History) tolerate higher compression rates than analytical reasoning tasks (College Medicine). QwQ-32B maintains perfect accuracy on History tasks while achieving

20.1% token reduction, whereas medical reasoning shows more sensitivity to aggressive compression. This suggests that different cognitive processes exhibit varying degrees of redundancy, opening avenues for adaptive, domain-aware compression strategies.

## H  EXTENDED DISCUSSION ON LIMITATIONS AND FUTURE WORK

Our framework provides a robust method for CoT compression, but it also highlights several important areas for future investigation.

**Sensitivity to Step Segmentation**  A core methodological choice in our work is the definition of a "reasoning step," which we segment based on the $(\n\n)$ delimiters generated by the LRM. This approach is practical and effective, but the entropy patterns we identify might be sensitive to this specific segmentation strategy. Different step boundaries—for instance, segmenting by sentences or logical clauses—could potentially alter the calculated entropies and the resulting compression performance. Future work should investigate the robustness of our findings to alternative segmentation methods and explore more semantically grounded techniques for automatically identifying discrete reasoning steps.

**Domain and Modality Generalizability**  Our validation focuses primarily on mathematical reasoning and select knowledge-based tasks from MMLU. While we demonstrate strong performance in these areas, the transferability of step entropy as a universal metric for redundancy is not yet established for all reasoning types. The nature of redundancy in more open-ended, creative, or commonsense reasoning tasks may differ significantly. Therefore, an important direction for future work is to apply and adapt our framework to a broader range of domains. Furthermore, extending this method to multimodal reasoning, where steps may involve interpreting images or data visualizations, represents a challenging but valuable next frontier.

**Adaptive and Model-Aware Compression**  Our experiments establish an 80% low-entropy pruning threshold that works remarkably well across several models. However, we acknowledge that this fixed threshold may not be universally optimal. As shown in our own experiments, different models or problem domains can exhibit unique redundancy patterns. This suggests a need for more dynamic approaches. A promising area for future work is the development of an adaptive compression strategy. Such a system could dynamically adjust the pruning ratio $\kappa$ based on the model's architecture, the specific problem's complexity, or the reasoning domain, moving from a static threshold to a more intelligent, context-aware compression policy.

## I  BROADER IMPACT

Our work addresses the critical challenge of reasoning efficiency in practical LLM deployment, where verbose Chain-of-Thought processes create significant computational bottlenecks. By providing a principled method for identifying and removing redundant reasoning steps, we enable more sustainable and accessible deployment of Large Reasoning Models. The interpretability benefits of maintaining explicit reasoning chains while achieving substantial compression offer advantages over latent reasoning approaches. Practitioners can retain the transparency and verifiability of explicit Chain-of-Thought while significantly reducing computational overhead. Our findings contribute to the theoretical understanding of reasoning structures in Large Language Models, revealing that current models generate substantial redundancy in their thought processes. This insight informs future model design and training methodologies, potentially leading to more efficient reasoning architectures.

