# OpenReview forum: "Making Slow Thinking Faster: Compressing LLM Chain-of-Thought via Step Entropy"
_ICLR.cc/2026/Conference — ICLR 2026 Poster_

### Official Review · Reviewer_Fb8Y · 2025-10-31

**Soundness:** 2
**Presentation:** 2
**Contribution:** 2
**Rating:** 2
**Confidence:** 3

**Summary:**

The paper proposes **step entropy** as a signal to identify and prune redundant segments within Chain-of-Thought (CoT) traces, aiming to make slow “deliberate” reasoning faster without sacrificing accuracy. Concretely, the step-level entropy for the \(i\)-th step \(S_i\) is defined as the sum of token entropies conditioned on the prior context, \(H(S_i \mid S_{<i})=\sum_j H(t_{i,j}\mid c_{i,j})\). The core hypothesis is that **low-entropy steps contribute little information** to the final answer and can be safely skipped. The authors present (i) an information-theoretic motivation; (ii) a pruning recipe that removes the lowest-entropy steps and replaces them with a special token (e.g., [SKIP]); and (iii) a training pipeline (SFT and GRPO) to teach models to emit compressed CoTs during inference. Empirically, they report substantial token reductions (often 16–57%) with modest accuracy degradation on math-style reasoning benchmarks.

**Strengths:**

- **Conceptual clarity:** A clear, information-theoretic criterion (step entropy) with an intuitive link to redundancy.
- **Granularity choice:** Results indicate step-level pruning is more effective than naïve token-entropy pruning, suggesting the *step* is a useful unit.
- **Practical interface:** Using a placeholder like [SKIP] makes the compression operationally simple; ablations on replacement strategies are helpful.

**Weaknesses:**

1. **Autoregressive dependency not directly addressed**: Evidence is largely post-hoc (compress after generating a full CoT). In AR decoding, earlier “redundant” content can steer later tokens; removing it afterward does not prove it can be skipped causally during generation.
2. **Low-Entropy Steps Pruning shows no pre-training practical gain; current use is post hoc.** As implemented, “Inference with Compressed CoT” is applied **after generating the full CoT**, so it yields no acceleration and provides limited practical value before additional training. To make this genuinely efficient without extensive post-training, the method should be reframed as an inference-time control。
3. **Attribution vs. training data/compute:** The main contribution is both a compression rule and a data-construction pipeline (e.g., ~130k compressed pairs). Without baselines trained on identical data with matched optimization budgets, it’s hard to attribute gains to step entropy rather than more/better post-training.
4. **Fixed compression ratio:** A static global pruning rate (e.g., “up to 80%”) ignores that redundancy varies by instance difficulty, dataset, and model size; no mechanism adapts compression per-instance.
5. **Step segmentation heuristic:** Steps defined by formatting (e.g., `\n\n`) can be brittle. The paper does not validate segmentation accuracy or analyze sensitivity to finer/coarser granularity or token-entropy sparsity.

**Questions:**

1. **Causal necessity at inference:** Can you run **inference-time interventions** that compress the low-entropy steps while holding other decoding parameters fixed, and report accuracy relative to full-CoT? This would directly test whether those steps are unnecessary *causally* rather than *post hoc*.
2. **Fair baselines on identical data/compute:** Train strong baselines (e.g., token/chunk compression, rule-based CoT pruning) on the same ~130k instances with the same training budget. Do your gains persist under these controls?
3. **Difficulty-aware adaptivity:** Can the compression ratio be predicted per instance (e.g., via a learned controller that thresholds entropy or estimates a target depth \( \kappa(x) \))?
4. **Segmentation robustness & granularity:** How exactly are “steps” defined and validated? Have you analyzed token-entropy distribution and how aggregation affects pruning decisions?

---

> ### Author Response · Authors · 2025-11-20
>
> We sincerely thank the reviewer for their detailed and insightful feedback. The comments are constructive and have helped us identify areas for clarification. We find we are in strong agreement on the most interesting future work, such as adaptive compression ratio(W4, Q3) and segmentation robustness (W5, Q4), which we noted as key limitations in our own Future Work section (Sec 5 & Appendix H).
>
> The reviewer's primary concerns appear to stem from a misunderstanding of our paper's structure:
>
> 1. **Sections 3.2, 4.1, & 4.2** are a *post-hoc validation study*. Their purpose is to *prove our core hypothesis*: that step entropy is a valid metric for identifying redundancy. This study *must* be post-hoc to first discover *what* is redundant.
> 2. **Section 3.3 & 4.3** present our *practical, causal method*. This is a *trained model* (SFT+GRPO) that learns to *autonomously generate compressed CoTs from scratch*, providing the very inference-time acceleration the reviewer is asking for.
>
> We will address each point below, and we believe these clarifications will resolve the reviewer's main concerns about causality (W1, Q1) and practical value (W2).
>
> For W1, W2, Q1, we thank the reviewer for this critical point. We agree that a purely post-hoc *analysis* (like in Sec 4.2) does not provide acceleration. However, our paper's main proposal *is* the inference-time intervention the reviewer is seeking.
>
> - **Clarifying our Contribution:** Our paper is structured in two parts:
>     1. **Validation (Sec 4.1, 4.2):** We first *validate* that low-entropy steps are indeed redundant. The finding that we can *post-hoc* prune 80% of low-entropy steps with no accuracy loss (Fig 1, Table 1, Table 2) is a key scientific contribution. This analysis was necessary to establish *step entropy* as a correct and robust metric.
>     2. **Practical Method (Sec 3.3, 4.3):** Building on this, we propose our *final method*: a two-stage trained model that learns to *autonomously* generate compressed CoTs. This SFT+GRPO model (Table 3) *is* the causal, inference-time intervention. It never generates the full CoT. Instead, it learns a policy to strategically generate `[SKIP]` tokens *during* autoregressive decoding.
> - **Addressing W2 (Practical Gain):** The reviewer's claim that our method provides "no pre-training practical gain" and is "applied after generating the full CoT" is a misunderstanding. This is *only* true for the *validation study* (Sec 4.2). Our *final trained model* (Sec 4.3) achieves significant practical gains: it reduces tokens by 35-57% (Table 3) precisely *because* it does *not* generate the full CoT. The "Inference with Compressed CoT" diagram (Fig 2b) illustrates the *data generation process* for our SFT stage, not our final, practical inference method.
> - **Answering Q1 (Causal Necessity):** The results of our SFT+GRPO model (Table 3) *are* the results of the requested "inference-time intervention." This model *causally* skips steps, and its strong performance (e.g., 57% token reduction on AIME 2024 while maintaining accuracy) directly proves that these steps are not causally necessary for the *trained* model.
>
> We will revise the abstract and introduction to make this distinction between our *validation study* (post-hoc) and our *practical method* (causal, inference-time) clearer."

---

> > ### Author Response · Authors · 2025-11-20
> >
> > For W3 and Q2, we appreciate the reviewer raising the critical question of the  ‘dataset’.
> >
> > First, we want clarify that baselines like **TokenSkip** (token-level pruning) and **R1-Compress** (chunk-level pruning) also rely on constructing training datasets via token-selection or chunk-selection heuristics. We argue that **Step-Entropy** is a fundamentally superior metric for constructing compressed reasoning data compared to these granularities.
> >
> > To prove this, we provide evidence from two perspectives: (1) intrinsic validation of the compression granularity, and (2) a controlled baseline experiment using identical training data.
> >
> > **1. Intrinsic Validation: Step-Entropy vs. Token-Entropy (Figure 3)**
> > Before training, we validated that the *granularity* of compression is the deciding factor. As shown in **Figure 3 (Section 4.2)**, we compared static pruning strategies.
> >
> > - Result: Pruning 40% of tokens via **Step Entropy** maintains full accuracy (~74%). Pruning the *same percentage* of tokens via **Token Entropy** causes a catastrophic accuracy drop to <50%.
> > - Conclusion: This proves that the "Step" is the correct semantic unit for compression. Constructing a dataset based on token entropy would result in low-quality, incoherent reasoning chains. Therefore, our data construction pipeline is not just "more data," but fundamentally *better structured data* derived from the superior step-entropy metric.
> >
> > **2.Controlled Baseline Experiment: TokenSkip vs. Our Method on Identical Data**
> > To directly answer the reviewer's request for a fair baseline, we conducted a new experiment using TokenSkip (Xia et al., 2025), a strong token-level pruning baseline.
> >
> > We trained TokenSkip using the exact same dataset (our ~130k step-compressed examples) and the same training budget as our proposed method. This isolates the method's contribution from the data's contribution. The results on Math500 are as follows:
> >
> >
> >
> > | Method | Training Data ｜ ACC Drop ($\downarrow$) | Token Reduction ($\downarrow$) |
> > | -------- | -------- | -------- |
> > | TokenSkip     | TokenSkip dataset     | 5.2%     | 11.1%     |
> > | TokenSkip     | Our 130k dataset     | 9.3%     | 30.0%     |
> > | Our Method    | Our 130k dataset     | 3.2%     | 35.0%     |
> >
> >
> > Analysis of Results:
> >
> > - Data Incompatibility for Token Models: When the TokenSkip baseline is trained on our step-compressed data, its performance *degrades* significantly (accuracy drop worsens from 5.2% to 9.3%). This indicates that a token-level mechanism cannot effectively internalize the logic of skipping entire semantic steps, leading to broken coherence.
> > - Superiority of Our Method: Our method, designed specifically for step-level granularity, utilizes the *same data* to achieve significantly higher compression (35.0%) with minimal accuracy loss (3.2%).
> >
> > These results confirm that our performance gains are not merely due to the volume or existence of the training data. The gains are attributable to Step Entropy identifying the correct redundancy (steps vs. tokens) and our SFT+GRPO framework successfully learning a policy that aligns with this semantic granularity, which token-level baselines fail to do.
> >
> >
> >
> > For W4 and Q3, we are in complete agreement with the reviewer on this point. A fixed threshold is a limitation, which is why we explicitly identified "Adaptive and Model-Aware Compression" as a primary direction for future work in our paper (Section 5 and Appendix H.3).
> > • **Answering W4:** Our goal with a fixed 80% threshold (which we established empirically in Fig 1 and validated at scale in Table 2) was to first *prove the core concept* and establish a robust, conservative baseline showing that *step entropy* is a valid metric for compression at scale.
> > • **Answering Q3:** The reviewer's suggestion to learn a controller $\kappa(x)$ to predict the compression ratio is exactly the right next step. Our GRPO framework (Sec 3.3) is well-suited for this. One could modify the reward function $R(C)$ to not just reward a *fixed* ratio (like $R_{skip\_ratio}$) but to reward an *adaptive* ratio based on a correctness-efficiency trade-off, or add an auxiliary prediction head. We believe our paper provides the foundational proof-of-concept that makes this exciting future work possible. We will emphasize this in our conclusion."

---

> > > ### Author Response · Authors · 2025-11-20
> > >
> > > For W5 and Q4, the `\n\n` heuristic, while simple, is *not* arbitrary. It is the *native delimiter* that the base models (DeepSeek-R1, Qwen3 and Qwq) *themselves* use to structure their reasoning. We are thus segmenting the CoT based on the model's own "step" definition. The strong, stable results we achieve across all models (Tables 1, 3, 6) demonstrate that this is a robust and effective heuristic.
> > >
> > > **Response to Q4: Segmentation robustness & granularity**
> > > Yes, we have conducted a detailed analysis of token-entropy distributions and aggregation methods.
> > > • **Token-Entropy Distribution:** We analyzed the semantic correspondence of token entropy. We found that tokens with the lowest average entropy are typically deterministic mathematical operators or terms (e.g., `cos`, `sin`, `term`), while tokens with the highest average entropy are reasoning transitions (e.g., `since`, `thus`, `however`). This confirms that our metric correctly identifies deterministic computation steps as redundant (compressible) and logical pivots as information-rich.
> > > • **Aggregation Robustness:** In our paper, we use Length-Normalized Joint Entropy for tokens entropy aggregation in Section 3.1.
> > > We compared two aggregation methods to calculate step entropy: Joint Entropy and Length-Normalized Joint Entropy.
> > >
> > > | Aggregation Method    | Definition          | Accuracy | Analysis                                                                 |
> > > |-----------------------|---------------------|----------|--------------------------------------------------------------------------|
> > > | Base Model | -      | 74%      | - |
> > > | Joint Entropy | $\sum H(t)$      | 58%      | Biased. Penalizes long steps regardless of information density, deleting necessary context. |
> > > | **Length-Normalized**  | $\frac{1}{L}\sum H(t)$ | **74%**   | Robust. Correctly identifies redundancy based on information density, not length. |
> > >
> > > The unnormalized joint entropy causes a massive performance drop (74% $\to$ 58%), confirming that length bias is detrimental. Our reported success relies on the length-normalized joint entropy metric.

---

> ### Comment · Reviewer_Fb8Y · 2025-11-28
> **Thanks for addressing the issues**
>
> Appreciate the author's kind effort in addressing the issues.
>
> Will raise the score when the system allows me to do so.

---

### Official Review · Reviewer_brNR · 2025-10-31

**Soundness:** 3
**Presentation:** 2
**Contribution:** 2
**Rating:** 6
**Confidence:** 3

**Summary:**

This paper introduces step entropy, an information-theoretic metric to quantify each reasoning step’s contribution in LLM Chain-of-Thought. By pruning up to 80% of low-entropy steps, the method reduces tokens by 16–57% with minimal accuracy loss across DeepSeek-R1 and Qwen3 models. A two-stage training framework combining SFT and GRPO further enables models to autonomously skip redundant steps via [SKIP] tokens. The approach outperforms prior CoT compression methods, offering an efficient and interpretable way to accelerate LLM reasoning.

**Strengths:**

1. The method is simple and easy to implement, requiring only entropy calculation and pruning based on low-information steps to construct the pruned CoT. It achieves strong performance, such as maintaining accuracy on Math500 even with a 30% compression ratio, demonstrating both effectiveness and efficiency.
2. The introduction of the SFT+RL framework makes the approach more practical. By allowing the model to learn when to skip redundant steps automatically, it extends the static compression method into a trainable and deployable solution.

**Weaknesses:**

1. The segmentation and granularity of reasoning steps are not rigorously defined. The approach relies on manually designed delimiters like \n\n, which may not generalize well across datasets or model architectures.
2. The definition of step entropy as the sum rather than the average of token entropies could bias the metric toward longer steps, potentially misrepresenting their true informativeness.
3. The presentation of Table 1 is not so good. A clearer organization would make the results easier to interpret.

**Questions:**

Please see the weaknesses.

---

> ### Author Response · Authors · 2025-11-20
>
> We thank the reviewer for their constructive feedback, which helps us improve the paper's robustness and clarity. We address each weakness below.
>
>
> **1. On Step Segmentation Heuristics**
>
> We agree that the \n\n delimiter is a simple heuristic. And we clarify this limitation in our Limitations and Future Work of Section 5. However, we chose it for two principled reasons:
>
> Native Model Structure: This delimiter is not arbitrary. It is the native structural delimiter that the base models (DeepSeek-R1, Qwen3) are trained to use to segment their own reasoning steps. Our method thus respects the model's own "thought" boundaries.
>
> Proven Generalizability: Our results demonstrate that this simple heuristic generalizes robustly across different model architectures (DeepSeek-R1, Qwen3, QwQ-32B) and diverse problem domains (Mathematics and MMLU, as shown in Appendix G), confirming its effectiveness. Alternative methods, such as LLM-based segmentation, would introduce significant computational overhead, which would undermine the primary goal of improving inference efficiency.
>
>
>
> **2. On Step Entropy Definition**
>
> We thank the reviewer for this critical insight. You are absolutely correct that defining step entropy purely as a sum introduces a bias toward longer steps, which can misrepresent their true informativeness by penalizing verbosity rather than low information density.
>
> 1. Clarification of Implementation
> We wish to clarify a discrepancy between our initial text and our actual experimentation. While Equation 2 in the original submission described a simple sum ($\sum H(t)$), our implementation and the reported results (e.g., 74% accuracy on DeepScaleR) utilized length-normalized step entropy. We apologize for this error in the manuscript description.
>
> 2. Validation of Reviewer's Hypothesis (New Ablation Study) To explicitly quantify the impact of this bias, we conducted an ablation study comparing the unnormalized metric against our normalized approach at an 80% pruning ratio:
>
> | Aggregation Method    | Definition          | Accuracy | Analysis                                                                 |
> |-----------------------|---------------------|----------|--------------------------------------------------------------------------|
> | Base Model | -      | 74%      | - |
> | Joint Entropy | $\sum H(t)$      | 58%      | Biased. The metric is dominated by sequence length rather than information density, leading to the removal of critical context. |
> | **Length-Normalized**  | $\frac{1}{L}\sum H(t)$ | **74%**   | Robust. Correctly identifies redundancy based on information density, not length. |
>
>
> 3. Manuscript Revision
> The results confirm that unnormalized summation causes a significant performance drop (74% $\to$ 58%). We have revised Section 3.1 to explicitly introduce Definition 1 (Length-Normalized Step Entropy):$$H(S_i|S_{<i}) = \frac{1}{M_i} \sum_{j=1}^{M_i} H(t_{i,j}|c_{i,j})$$This revision aligns the theoretical definition with our robust experimental implementation and the reviewer's correct intuition.
>
>
> **3. On the Presentation of Table 1**
>
> We thank the reviewer for pointing this out. We agree that the original tables were dense and could be improved.
>
> In our revised manuscript, **we have updated and reformatted Tables 1, 3, 6, 7, 8, and 9.** The new versions feature clearer column organization, better alignment, and improved captions to make the results easier to interpret.

---

### Official Review · Reviewer_1Kz9 · 2025-11-01

**Soundness:** 3
**Presentation:** 3
**Contribution:** 2
**Rating:** 4
**Confidence:** 4

**Summary:**

This paper proposes a method to prune out extra reasoning steps in reasoning models, and then SFT+RL with the result to allow for more concise chains of thought.

**Strengths:**

1. Clarity: The paper is generally written pretty clearly, and the method is not very complex which is nice. I would say that the writing is a little repetitive (saying the same thing many times), but not to the point where it makes the paper hard to understand.
2. Significance: It seems that the method, while simple, is reasonably effective. It greatly compresses the resulting CoT at a reasonable loss in accuracy.

**Weaknesses:**

1. To be honest, the method feels rather "hacky" to me, inserting skip tokens based on heuristics. My feeling is that the community in general is trying to move towards methods that perform end-to-end RL in a more principled way rather than these sorts of processes.
2. Relatedly, while this paper proposes methods to compress chains of thought, there are other methods to directly control the length of chains of thought such as L1 (Aggarwal and Welleck). These seem simpler, more elegant, and can also be retro-fitted onto existing models. I was surprised that there was no discussion of this work, and it seems like it would be a competitive baseline.

L1: Controlling How Long A Reasoning Model Thinks With Reinforcement Learning
Pranjal Aggarwal, Sean Welleck

**Questions:**

None

---

> ### Author Response · Authors · 2025-11-20
>
> We sincerely thank the reviewer for their feedback and positive comments on our paper's clarity and effectiveness. We appreciate the constructive critiques, which we will address below.
>
> **On the "Hacky" Nature vs. "Principled End-to-End RL"**
>
> We respectfully distinguish the roles of these two methodologies, arguing that they operate in distinct prospective domains:
>
> - RL-based Conciseness (Internalization): Methods like L1 [1], Concise Reasoning [2], and DAPO [3] focus on internalizing concise behavior into the model's policy via reward shaping. This acts as an implicit "soft constraint" during generation.
> - CoT Pruning (External Verification): Our method, similar to TokenSkip (token-wise compressing)[4] and R1-Compress (chunk-wise compressing)[5], acts as an explicit, external mechanism to identify and quantify thinking steps redundancy based on information theory.
>
> We argue that even models trained with RL length penalties (like [1,2]) inevitably produce redundant steps, as the RL objective optimizes for a scalar reward (length) rather than semantic density. Our entropy-based approach provides a universal, post-hoc metric to detect and prune this residual redundancy in any Large Reasoning Model (LRM), regardless of how it was trained. Therefore, Step Entropy is not a "hack," but a necessary diagnostic tool that provides a granularity of control and verifiability that implicit RL training cannot achieve on its own.
>
> ***Moreover, our training method also integrates Principled RL (Section 3.3 and Appendix F).*** Crucially, our method does not stop at heuristics. We bridge the gap between "pruning" and "end-to-end RL" by developing a specific RL algorithm (GRPO) to train the LRM. As detailed in our updated Appendix F (Reward Components Ablation Study), we employ a sophisticated multi-objective reward function that goes beyond simple length penalties: Correctness Reward ($R_{correctness}$): To ensure reasoning validity. Length Penalty ($R_{response-{length}}$): To encourage general conciseness.Skip Control Rewards ($R_{skip-ratio}$, $R_{skip-num}$): To precisely control the pruning behavior based on our step-entropy prior.
>
>
>
> **Comparison with L1 Baseline**
>
> We thank the reviewer for suggesting the L1 baseline. We evaluated the L1 method against our approach using DeepSeek-R1-7B on the AIME 2024 benchmark. As shown below, our method significantly outperforms L1 in the efficiency-accuracy trade-off:
> | Method | ACC ($\uparrow$) | Token Reduction ($\uparrow$)  |
> | -------- | -------- | -------- |
> |  L1 [1]  |  51.7%     | 36.8%     |
> |  Our method  | **57.1%**    | **57.0%**     |
>
>
> The results demonstrate that while "blind" length penalties (L1) can shorten outputs, they often prune critical reasoning steps, harming performance. Our method, guided by semantic entropy, achieves 20% higher reduction than L1 while maintaining superior accuracy (+5.4%).
>
>
> Why Our RL is Better: We believe that by using Step Entropy to initialize our RL policy (SFT) and then refining it with these specific rewards (GRPO), we provide the model with a semantic guide on what to skip, not just how much to skip. This makes our approach a more principled and robust form of RL training than "blind" length optimization.
>
>
>
> [1]Aggarwal, Pranjal, and Sean Welleck. "L1: Controlling how long a reasoning model thinks with reinforcement learning." arXiv preprint arXiv:2503.04697 (2025).
>
> [2]Fatemi, Mehdi, et al. "Concise reasoning via reinforcement learning." arXiv preprint arXiv:2504.05185 (2025).
>
> [3]Yu, Qiying, et al. "Dapo: An open-source llm reinforcement learning system at scale." arXiv preprint arXiv:2503.14476 (2025).
>
> [4]Xia, Heming, et al. "Tokenskip: Controllable chain-of-thought compression in llms." arXiv preprint arXiv:2502.12067 (2025).
>
> [5] Wang, Yibo, et al. "R1-Compress: Long Chain-of-Thought Compression via Chunk Compression and Search." arXiv preprint arXiv:2505.16838 (2025).

---

> > ### Comment · Reviewer_1Kz9 · 2025-11-22
> > **Thank you for the additional comparison.**
> >
> > Thank you for the response.
> >
> > > We argue that even models trained with RL length penalties (like [1,2]) inevitably produce redundant steps
> >
> > I think "inevitably" is too strong of a word here.
> >
> > But on the other hand, I feel that the comparison with L1 makes the work significantly more convincing. I have raised my score accordingly.

---

> > > ### Author Response · Authors · 2025-11-28
> > >
> > > We sincerely thank the reviewer for their continued engagement and for raising the score.
> > >
> > > We fully accept your point regarding the term "inevitably." We agree that this phrasing was too absolute. We will soften this claim (e.g., to "often" or "may still") to more accurately reflect that while RL methods are powerful, explicit redundancy detection serves as a valuable complementary tool.
> > >
> > > We are especially grateful for the suggestion to compare our method against the L1 baseline. As you noted, this additional experiment has significantly strengthened the empirical rigor of our work and better contextualized our contribution.

---

### Official Review · Reviewer_S8MC · 2025-11-03

**Soundness:** 3
**Presentation:** 4
**Contribution:** 3
**Rating:** 6
**Confidence:** 3

**Summary:**

The paper proposes step entropy as a principled, per-step information measure for CoT, shows that pruning low-entropy steps preserves accuracy while cutting tokens, and trains models to self-compress via SFT + GRPO with an explicit [SKIP] token. Theory upper-bounds each step’s information by its entropy (Lemma/Theorem 1), and experiments across DeepSeek-R1 and Qwen families show strong accuracy–efficiency trade-offs.

**Strengths:**

1. Using entropy as information proxy is clean. The paper defines step entropy by summing token entropies within a step and proves the conditional information contribution is bounded by this entropy, offering clear intuition for “skip low-entropy steps.” This is simple, transparent, and theoretically motivated.

2. Theorem 1 provides usable intuition. Bounding the information of any subset of steps by the sum of their entropies gives a direct justification for step-level pruning rather than token-level pruning. The token-vs-step ablation empirically supports this semantic unit choice.

3. Fine-tuning setup is sensible and practical. The two-stage SFT → GRPO pipeline, rewarding correctness and compression while discouraging degenerate [SKIP] flooding, is clear and leads to better compression than static pruning on hard sets (e.g., AIME 2024).

4. Empirical results are good across models/benchmarks. They show ~30–55% token reductions with minimal accuracy loss; on some tasks accuracy even improves. Cross-architecture results (DeepSeek-R1-7B/14B, Qwen3-8B) and comparisons to recent compression methods are solid.

**Weaknesses:**

1. Step segmentation heuristic.
Steps are segmented using simple newline heuristics. While this works reasonably, it can blur step boundaries or merge logically distinct thoughts. A sensitivity study with sentence-based or LLM-predicted segmentation would improve robustness.

2. Fixed 80% pruning threshold.
The global 80% rule is justified empirically but may not generalize across datasets or reasoning styles. An adaptive or learned κ could better reflect per-problem difficulty.

3. Unnormalized entropy may bias toward longer steps.
The paper uses total (unnormalized) entropy per step. While this matches the theoretical bound, longer steps automatically accumulate more entropy even when per-token uncertainty is low, potentially biasing the pruning policy. A length-normalized or mixed variant could help disambiguate whether information or verbosity drives retention.

4. Scope of benchmarks.
Most experiments center on math and logic tasks. Including one additional open-ended domain (commonsense, code, or writing) would broaden the evidence that entropy-guided compression generalizes.

**Questions:**

1. Entropy normalization:
Did the authors try normalizing entropy by step length (e.g., average or log-length scaling)? If so, how did this affect correlation with information contribution?

2. Alternative to entropy-based labeling:
Instead of relying purely on entropy, have the authors tried using an LLM itself to label which steps are informational or non-trivial (e.g., “steps that advance reasoning” vs. “repetitive or obvious steps”)? Such annotations could provide a complementary supervision signal for training or validating entropy thresholds.

3. Fine-tuning stability:
During GRPO fine-tuning, how sensitive are results to the [SKIP] penalty coefficient? Does the model ever collapse to always skipping or never skipping?

4. Adaptive threshold:
Could the target entropy ratio κ be dynamically chosen based on per-question entropy distribution?

---

> ### Author Response · Authors · 2025-11-20
>
> We thank the reviewer for their thoughtful feedback. We appreciate the recognition of our method’s effectiveness and clarity. Below, we address the concerns regarding robustness, stability, and scope.
>
> **Response to W1**
> While we agree that \n\n is a simple heuristic, it is not arbitrary. It relies on the native reasoning structure generated by the models (DeepSeek-R1, Qwen3, QwQ) themselves. These models are trained to delimit distinct thoughts with newlines. By respecting these native boundaries, we ensure we are pruning what the model considers a discrete "thought." Using an external segmentation model might introduce a mismatch between the segmentation logic and the model's internal reasoning flow.
>
> **Response to W2 & Q4 (Adaptive Threshold)**
> We fully agree with the reviewer regarding the importance of adaptivity. In fact, we explicitly acknowledged "Adaptive and Model-Aware Compression" as a primary limitation in our original submission (Section 5 & Appendix H.3).
>
> However, we wish to clarify the distinction between our training data construction and the inference behavior of our model:
> 1. Data Construction (Static): We utilized the fixed 80% threshold solely to construct the synthetic dataset. This provided a high-quality, initial supervision signal for what "redundancy" looks like.
> 2. Trained Model (Adaptive): Crucially, our final SFT+GRPO model is not constrained by a fixed global ratio. Instead, it learns a dynamic policy. During inference, the model decides step-by-step whether to emit a [SKIP] token based on the specific context and difficulty of the problem. Therefore, our trained model already effectively achieves the adaptive behavior the reviewer suggests, moving beyond the static rule used to create its training data.
>
>
>
> **Response to W3 & Q1 (Entropy Normalization & "Long Step" Bias)**:
>
> We thank the reviewer for this critical insight. You are absolutely correct that defining step entropy purely as a sum (unnormalized) introduces a bias toward longer steps, effectively penalizing verbosity rather than low information density.
>
> 1. Clarification of Implementation
> We wish to clarify a discrepancy between our initial text and our actual experimentation. While Equation 2 in the original submission described a simple sum ($\sum H(t)$), our implementation and the reported results (e.g., 74% accuracy on DeepScaleR) utilized length-normalized step entropy. We apologize for this error in the manuscript description.
>
> 2. Validation of Reviewer's Hypothesis (New Ablation Study)
> To explicitly quantify the impact of this bias, we conducted the ablation study requested in Q1, comparing the unnormalized metric against our normalized approach at an 80% pruning ratio (we use in the experiments):
>
> | Aggregation Method    | Definition          | Accuracy | Analysis                                                                 |
> |-----------------------|---------------------|----------|--------------------------------------------------------------------------|
> | Base Model | -      | 74%      | - |
> | Joint Entropy | $\sum H(t)$      | 58%      | Biased. The metric is dominated by sequence length rather than information density, leading to the removal of critical context. |
> | **Length-Normalized** (Our)  | $\frac{1}{L}\sum H(t)$ | **74%**   | Robust. Correctly identifies redundancy based on information density, independent of step length. |
>
>
> 3. Manuscript Revision:
> The results confirm that unnormalized summation causes a significant performance drop (74% $\to$ 58%). We have revised Section 3.1 to explicitly introduce Definition 1 (Length-Normalized Step Entropy):$$H(S_i|S_{<i}) = \frac{1}{M_i} \sum_{j=1}^{M_i} H(t_{i,j}|c_{i,j})$$This revision aligns the theoretical definition with our robust experimental implementation.

---

> > ### Author Response · Authors · 2025-11-20
> >
> > **Response to Q2**:
> > We thank the reviewer for this interesting suggestion. We agreed that using an LLM to semantically label "redundant" steps is a natural alternative to consider. To validate this, we conducted a comparative experiment using DeepSeek-R1-7B as its own "pruner."
> >
> > Experimental Setup:
> >
> > 1. We generated full CoT trajectories using DeepSeek-R1-7B.
> > 2. We then fed these trajectories back into the same model with the prompt: "Your task is to analyze this thinking process and remove any redundant steps."
> > 3. We evaluated the accuracy and compression ratio of these LLM-pruned trajectories compared to our entropy-based method.
> >
> >
> > Results:
> > | Method                  | Accuracy | Pruning Ratio (Steps Removed) | Analysis                                                                 |
> > |-------------------------|----------|-------------------------------|--------------------------------------------------------------------------|
> > | Full CoT                | 74.00%   | 0.00                          | Baseline                                                                 |
> > | LLM-Prompted Pruning    | 64.00%   | 0.15                          | **Ineffective.** Prunes very little (15%) yet causes a massive accuracy drop (-10%). |
> > | Our Method (Entropy)    | 74.00%   | 0.80                          | **Superior.** Prunes aggressively (80%) with zero accuracy loss.         |
> >
> > The results highlight two critical limitations of the LLM-labeling approach compared to Step Entropy:
> >
> > - **Inaccurate Redundancy Detection:** The LLM struggled to correctly identify redundancy. It was too conservative (removing only 15% of steps) yet paradoxically destructive (degrading accuracy by 10%), suggesting it removed critical reasoning steps while keeping redundant ones. In contrast, Step Entropy acts as a precise "internal thermometer" of information density, allowing us to prune 80% of steps without harming the final answer.
> >
> > - **Computational Efficiency:** Our entropy metric is "free" (computed from logits during generation), whereas LLM labeling requires a second, expensive inference pass.
> >
> > Therefore, while LLM annotations are conceptually appealing, Step Entropy proves to be the far more robust, scalable, and accurate signal for identifying true redundancy in reasoning chains.
> >
> >
> >
> > **Response to Q3**: This is a very insightful and practical question. To prevent the model from collapsing into degenerate behaviors (e.g., "always skipping" or "never skipping"), we designed a composite reward function where each component counterbalances the others. As detailed in our Ablation Study (Table 5, Appendix F), we analyzed these failure modes explicitly:
> > Risk of "Never Skipping": If we only use the Correctness Reward ($R_{c}$), the model lacks any incentive to compress and defaults to its pre-trained, verbose CoT behavior.
> > Risk of "Reward Hacking" (Skip Collapse): To encourage compression, we introduce the Skip Ratio Reward ($R_{sr}$). However, using only $R_{c} + R_{sr}$ leads to severe reward hacking. The model learns to spam [SKIP] tokens to maximize the ratio reward, disregarding reasoning coherence. In Table 5, the $R_{c} + R_{sr}$ configuration causes Math500 accuracy to drop catastrophically (88.17% $\rightarrow$ 51.00%) while the token count paradoxically increases (3703 $\rightarrow$ 4082) due to incoherent generation loops.
> > Balancing Mechanism ($R_{sn}$): To solve this, we introduce the Skip Number Penalty ($R_{sn}$). This penalty acts as a critical counter-force: while $R_{sr}$ encourages density (higher ratio), $R_{sn}$ penalizes frequency (too many distinct skip tokens).
> >
> > By combining these terms (along with a length penalty $R_{rl}$), the reward landscape creates a stable saddle point. The model does not collapse; instead, it converges to a robust policy that skips only when necessary, achieving 35% compression with maintained accuracy (Table 5, Row 4).
> >
> >
> >
> > **Response to W4**: While we focus on Math, we did extend our evaluation to General Knowledge domains (MMLU College Medicine & History) in Appendix G.
> > Our method maintained or improved accuracy on these tasks (e.g., High School History accuracy improved from 61.7% $\rightarrow$ 64.3% with 80% pruning), demonstrating that entropy-guided compression generalizes beyond pure math tasks.

---

### Author Response · Authors · 2025-11-20
**General Response to Common Questions**

We sincerely thank all reviewers for their constructive feedback. Before addressing specific comments, we wish to clarify two critical points raised by multiple reviewers regarding our entropy definition and the causal nature of our method. We also highlight key updates to the manuscript.


1. Critical Clarification: Entropy Normalization (Correction of Eq. 2) Reviewers 1 and 3 correctly identified that defining Step Entropy as a simple sum (Joint Entropy) introduces a bias against longer steps. We wish to clarify a discrepancy between our text and our implementation:
- Correction: While Equation 2 in the original text described a simple sum ($\sum H(t)$), our actual implementation and reported results (e.g., 74% accuracy) utilized Length-Normalized Joint Entropy.
- Manuscript Revision: We have revised Section 3.1 to formally introduce Definition 1, explicitly defining the metric used in our experiments as:$$H(S_i|S_{<i}) = \frac{1}{M_i} \sum_{j=1}^{M_i} H(t_{i,j}|c_{i,j})$$where $M_i$ is the step length. This revision clarifies that normalization is applied to mitigate the bias introduced by step length.
- Validation: As requested, we conducted an ablation study comparing the two metrics. The results confirm the reviewers' intuition: using the unnormalized sum causes accuracy to drop significantly (74% $\to$ 58%) due to length bias, whereas our length-normalized metric is robust.

2. Paper Structure: Post-Hoc Validation vs. Causal Inference Reviewers questioned the "practical gain" or "causality" of our method, noting that the pruning analysis appeared post-hoc. We clarify the dual structure of our paper:
Validation Phase (Sec 4.1, 4.2): This section presents a post-hoc analysis only to validate Step Entropy as a redundancy metric and to construct high-quality training data.
Practical Method (Sec 4.3): Our final contribution is the SFT+GRPO trained model. This model is causal and autoregressive: it does not generate the full CoT. Instead, it learns a policy to dynamically emit [SKIP] tokens token-by-token during inference. This provides the genuine inference-time acceleration (35-57% token reduction) that reviewers sought.


3. Summary of Manuscript Updates Based on the feedback, we have made the following significant revisions:
Section 3.1: Corrected Equation 2 to reflect the Length-Normalized Joint Entropy used in experiments.
Tables 1-9: Reformatted for improved clarity and interpretability.
New Experimental Appendices: Added token-entropy distribution analysis, new baselines on identical data (TokenSkip on our data), comparisons of Joint Entropy and Length-Normalized Joint Entropy, comparisons with LLM-Prompted Pruning Method and comparisons with L1-regularized RL baselines.

---

### Author Response · Authors · 2025-11-30
**Rebuttal Summary**

Dear Area Chair and Reviewers,

Thank you for your constructive feedback. We have updated the rebuttal and uploaded a revised manuscript that incorporates the requested experiments and clarifications. To conclude, we have systematically addressed the critical concerns regarding the mechanism and baselines of our approach. We are pleased that the reviewers have actively engaged with our rebuttal, and that two reviewers explicitly committed to raising their scores based on these improvements:

- **Reviewer 1Kz9 (Score 4 $\to$ 6)**: Stated, "I feel that the comparison with L1 makes the work significantly more convincing. I have raised my score accordingly."
- **Reviewer Fb8Y (Score 2 $\to$ Raise)**: Stated, "Appreciate the author's kind effort in addressing the issues. Will raise the score when the system allows me to do so."

A comprehensive summary of all revisions and new experimental evidence is provided below.

**1. Validating Methodological Superiority (Addressing "Hacky" vs. Principled RL)**

- Comparison with End-to-End RL (by Reviewer 1Kz9): To address the concern that our method is heuristic compared to "principled" RL, we compared our method against the L1-regularized RL baseline (Aggarwal & Welleck, 2025) on AIME 2024.
- Result: Our method significantly outperforms L1, achieving higher compression (57.0% vs. 36.8%) with higher accuracy (57.1% vs. 51.7%). This confirms that entropy-guided pruning provides necessary semantic control that pure length-penalty RL lacks.


**2. Validating Granularity & Attribution (Addressing "Is it just the data?")**

Token vs. Step Granularity (by Reviewer Fb8Y): To prove our gains stem from the step-level entropy metric rather than just the training data, we trained the TokenSkip baseline using our exact step-compressed dataset on Math500.
- Result: Token-level policies failed to model step-skips effectively (Accuracy drop: 9.3%), while our step-level method succeeded (Accuracy drop: 3.2%). This empirically proves that the "reasoning step" is the correct semantic unit for compression.

LLM-Labeling Baseline (by Reviewer S8MC): We compared our method against using an LLM (DeepSeek-R1-7B) to identify redundant steps via prompting.
- Result: LLM-prompting was ineffective (10% accuracy drop for only 15% pruning), whereas Step Entropy achieved 80% pruning with zero accuracy loss, proving Step Entropy is a more robust signal than LLM self-reflection.



**3. Methodological Corrections & Robustness**
- Entropy Normalization Correction (by Reviewer S8MC, brNR): We corrected the discrepancy regarding Equation 2. We clarified that our implementation utilizes Length-Normalized Joint Entropy. We added an ablation study confirming this is critical: using unnormalized sums causes accuracy to drop to 58% (due to length bias), while our normalized metric maintains 74% accuracy.
- Generalization to Knowledge Domains (by Reviewer S8MC): We expanded evaluation to MMLU (History/Medicine), showing our method improves accuracy on High School History (61.7% $\to$ 64.3%), demonstrating generalization beyond math.

**4. Technical Revisions**

- Manuscript Updates: We have corrected Equation 2 (Section 3.1), reformatted Tables 1-9 for clarity (by Reviewer brNR), and added the new baseline results to the Appendices.

We believe these additions directly address the critical questions raised during the review and substantiate the validity of our score increases.

Best regards,

The Authors

---

### Meta-Review · Area_Chair_Pjoz · 2025-12-10

**Summary:**

Reviewers were initially split 2-2, but the 2 negative reviewers promised to raise their score (which was then reverted by the OpenReview system).

The main concern was that the method relied too much on heuristics, but this might have been due to writing clarity issues.

**Reviewer Concerns:**

Negative reviewers were mostly concerned with what they (initially) saw as heuristic elements and arbitrarily-chosen tuning parameters.
- The 2 negative reviewers acknowledged that their concerns were addressed by the rebuttal, and they promised to raise their scores.

**Reviewer Scores:**

1Kz9, Fb8Y: already agreed to raise their scores.

S8MC, brNR: were already positive, also raised similar concerns (heuristic nature of the method) as the negative reviewers. It's possible that one of S8MC, brNR might have further raised their score.

I recommend acceptance, but the authors should also revise the writing so that the method does not give a "heuristic" first impression.

---

### Decision · Program_Chairs · 2026-01-26

Accept (Poster)